# Development of an automatic thresholding method for wake meandering studies and its application on the data set from scanning wind lidar

Maria Krutova[1], Mostafa Bakhoday-Paskyabi[1], Joachim Reuder[1], and Finn Gunnar Nielsen[1]

[1]Geophysical institute and Bergen Offshore Wind Centre, University of Bergen, Allégaten 70, 5007 Bergen, Norway

**Correspondence:** Maria Krutova (maria.krutova@uib.no), Mostafa Bakhoday-Paskyabi (mostafa.bakhoday-paskyabi@uib.no)

**Abstract.**

Wake meandering studies require knowledge of the instantaneous wake evolution. Scanning lidar data are used to identify the wind flow behind offshore wind turbines but do not immediately reveal the wake edges and centerline. The precise wake identification helps to build models predicting wake behavior. The conventional Gaussian fit methods are reliable in the near-wake area but lose precision with the distance from the rotor and require good data resolution for an accurate fit. The thresholding methods, i.e., selection of a threshold that splits the data into background flow and wake, usually imply a fixed value or manual estimation, which hinders the wake identification on a large data set. We propose an automatic thresholding method for the wake shape and centerline detection, which is less dependent on the data resolution and quality and can also be applied to the image data.

We show that the method performs reasonably well on large-eddy simulation data and apply it to the data set containing lidar measurements of the two wakes. Along with the wake identification, we use image processing statistics, such as entropy analysis, to filter and classify lidar scans.

The automatic thresholding method and the subsequent centerline search algorithm are developed to reduce dependency on the supplementary data such as free-flow wind speed and direction. We focus on the technical aspect of the method and show that the wake shape and centerline found from the thresholded data are in a good agreement with the manually detected centerline and the Gaussian fit method. We also briefly discuss a potential application of the method to separate the near and far wakes and to estimate the wake direction.

## 1 Introduction

A wake is a complex dynamic structure forming behind a wind turbine due to the kinetic energy extraction from the incoming wind flow. The wake region is characterized by decreased wind speed and increased turbulence intensity. The relative velocity deficit, or wake deficit, is strongest right after the wind turbine. Strongly affected by wind turbine rotor, the region extends up to $4-5$ rotor diameters depending on the terrain characteristics and stability conditions (Stevens and Meneveau, 2017; Porté-Agel et al., 2020). The wake transitions to the far wake, where the recovery to the free flow is considerably slowed down; at the

same time, the wake width increases up to three rotor diameters according to observations (Aitken et al., 2014). The turbine spacing in operational wind farms usually reaches $7-10D$ (e.g., London Array), although the optimal spacing is estimated to be even higher in order to reduce the wake effect on downstream turbines (Meyers and Meneveau, 2012; Stevens, 2016). Since the generated wind power is proportional to the cube of the wind speed $U^3$, the power production gradually decreases if the incoming wind speed drops below the rated wind speed. The increased turbulence intensity negatively affects the turbine fatigue loads (Lee et al., 2012). Studying the wake behavior is hence crucial to estimating both the actual power production and the overall lifetime of a wind farm.

Not only the wake expands, but it is also subjected to the wake meandering – oscillations along the rotor axis caused by the movement of large eddies (Larsen et al., 2007, 2008). While the near wake remains primarily stable and follows the wind direction, the far wake oscillates randomly in the horizontal plane with an amplitude exceeding $0.5D$ (Howard et al., 2015; Foti et al., 2016). The far wake also oscillates in the vertical plane, although the velocity fluctuations there are weaker (España et al., 2011). As a result, a downstream turbine is exposed to intermittent flow and, consequently, unequal fatigue loads (Muller et al., 2015; Moens et al., 2019). Additionally, the wake in the Northern hemisphere slightly turns clockwise due to the Coriolis effect (Abkar and Porté-Agel, 2016; van der Laan and Sørensen, 2017) adding more complexity to the wake evolution over time. Knowing only the velocity deficit at a certain downstream distance is insufficient, since the wake meandering strength is characterized by the standard deviation of the wake center. Therefore, the wake meandering analysis requires the knowledge of the wake centerline to quantify the instantaneous wake effect on the downwind structures. An appropriate detection method should be able to perform wake identification by separating the wake from the free flow and wake characterization by estimating the wake centerline and its statistical characteristics (Quon et al., 2020). Methods' application and capabilities are highly dependent on the input data available.

Measurement campaigns that use scanning lidars provide the most relevant data on the wind flow in a particular wind farm (Trujillo et al., 2010, 2011; Herges et al., 2017). Due to the technical restrictions and cost of lidar installation, it is complicated to obtain a three-dimensional scan of the flow around the whole wind farm, although the flow can be reconstructed for a single turbine (Beck and Kühn, 2019). Still, the measurement campaigns span along several months and require data preprocessing to sort out invalid measurements. A controlled experiment can be performed on a wind tunnel for model validation or reproduction of specific flow conditions (Snel et al., 2007; Chamorro and Porté-Agel, 2010). The particle image velocimetry (PIV) provides good spatial and temporal resolution of the measured wind field but deals with the scaled models and has to account for their limitations. A different approach is running a large-eddy simulation (LES) of a wind turbine or a wind farm. While LES provides a wide range of possibilities to simulate atmospheric conditions and wind farm configuration, its representation of a wake strongly depends on the implemented turbulence closure (Moriarty et al., 2014; Mehta et al., 2014; Martínez-Tossas et al., 2018) and wind turbine model (Porté-Agel et al., 2011; Martínez-Tossas et al., 2015). A relatively new development is quantitative study of wind farm wakes from the satellite data (Ahsbahs et al., 2020). The satellites generally have a lower spatial resolution than scanning lidars and measure wind speed on the horizontal near-surface plane but still provide general information on the flow around wind farms.

Several wake identification methods exist, varying in complexity and input data requirements (Quon et al., 2020). Among the variety of the methods, we focus on the thresholding and Gaussian fitting because they are applicable to a 2D lidar scan in a horizontal or inclined plane. The most common wake identification method is to fit a one- or two-dimensional Gaussian distribution to the velocity deficit across the wake at various downstream positions and get estimations of the wake center and width from the fitted function (Fleming et al., 2014; Vollmer et al., 2016; Krishnamurthy et al., 2017). The method can be applied both to the averaged and instantaneous wake, although the irregular wake shape of the latter complicates the detection. For better accuracy, the fitting requires wind speed data in a fine spatial resolution. A sufficient spatial resolution is achieved by large-eddy simulation or particle image velocimetry. The Gaussian fit method can also be applied to the scanning wind lidar data, provided the wake region is resolved well enough. Overall, the fitting method efficiency depends on the data quality and spatial resolution. The method also requires the free-flow wind speed to calculate the wake deficit.

Alternatively, a threshold value can be defined. In the simplest case, the threshold splits the range of available values into two: all values below the threshold fall into one group, while the remaining values form the second group. When applied to the wind field for the wake identification, the threshold would split the data into the wake and free-flow points. Thresholding methods depend less on the data resolution and mainly rely on the wind speed values. The simplest thresholding method sets a threshold based on the wind speed ratio in the wake and the free flow. As shown by España et al. (2011), the method is effective for a regular flows, e.g., in a wind turbine: a threshold of 95% of the free-flow wind speed identified the continuous part of the wake up to the downstream distance of $6-8D$. The resulting shape required smoothing and filtering to reduce the noise. Bastine et al. (2015) used a stricter threshold of 40% of the maximum wake deficit on the LES data to extract the wake core and perform proper orthogonal decomposition on the processed wind field.

The thresholding method is not widely used due to its restriction: it applies an empirical coefficient that does not account for the data quality and wind speed fluctuations in the flow field, which may be a common issue for a lidar scan. We propose an automated threshold estimation, previously developed for the whitecaps detection – Adaptive Thresholding Segmentation method (ATS) (Bakhoday-Paskyabi et al., 2016). We adapt the method for the wake identification and develop new routines to estimate the wake centerline without a priori knowledge of the wind direction.

This study focuses on the technical aspect of the ATS method and discusses its advantages and limitations. The method is applied to a scanning lidar data set containing wakes from two wind turbines and various wake-wake interactions. The measurements and LES setup are described in Sect. 2. Lidar data required additional preprocessing, described in Sect. 3. In the same section, we preview diagnostic techniques by using image entropy to evaluate and classify the data. The application of the image processing method to the wake identification and characterization is detailed in Sect. 4. We demonstrate our algorithm on the idealized LES data as a proof of concept in Sect. 5. We then apply the same algorithm to the lidar data and compare the result with the manual wake detection, deficit-based thresholding and the Gaussian fit method in Sect. 6. The findings are summarized in Sect. 7. In the Appendix, we briefly discuss the differences between wake identification from the lidar scan data and the respective grayscale image.

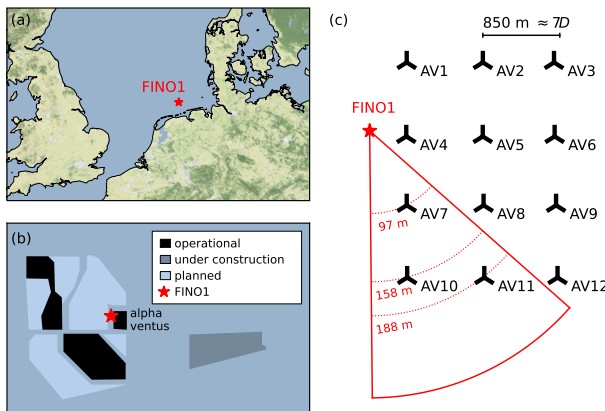

**Figure 1.** A schematic shows (a) the location of FINO1 platform, map made with Natural Earth, (b) wind farms and platforms near FINO1, status in 2015-2016, and (c) *alpha ventus* wind farm layout, maximum lidar scan area and scanning height at the position of each wind turbine.

## 2 Data description

### 2.1 Lidar and reference data

We use measurements of wind speed and wind direction recorded during the Offshore Boundary-Layer Experiment at FINO1 (OBLEX-F1) campaign. The FINO1 platform is located in the North Sea at 54° 00' 53.5" N 6° 35' 15.5" E, 45 km to the north of the German island of Borkum. The *alpha ventus* wind farm is located in the vicinity of FINO1 and consists of 12 wind turbines arranged in a rectangular pattern (Fig. 1). The wind turbines AV1−AV6 are of the type Repower 5M with a hub height of 92 m and a rotor diameter of 126 m; AV7−AV12 are of the type AREVA M5000 with a hub height of 91.5 m and a rotor diameter $D$ of 116 m. The row and column distances between the turbines vary within 800−850 m, approximately seven rotor diameters, $7D$. The distance between FINO1 and the closest wind turbine, AV4, is 405 m.

The FINO1 meteorological mast has a cup anemometer installed at 90 m above sea level and a vane installed at 100 m above sea level. The wind speed and direction measured with those instruments are used to characterize the free flow. We will further refer to them as the reference wind speed and direction, respectively.

The scanning Doppler wind lidar Leosphere WindCube 100S installed at FINO1 is oriented towards the *alpha ventus* wind farm. The closest scanned wind turbine, AV7, is located at 919 m or $7.92D$ from FINO1 (Fig. 1c). The lidar is installed at 23.5 m above sea level and operates in a Plan Position Indicator (PPI) scanning mode. In this mode, the azimuth of the lidar beam changes between 131.5° and 179.5° at an elevation angle of 4.62°. The lidar scans the south-western sector of the *alpha ventus* wind farm and captures wake patterns from two wind turbines, AV7 and AV10. The third wind turbine, AV11, stays outside of the lidar range in most scans, but a part of its wake is visible for the specific wind directions. The wind turbine AV7

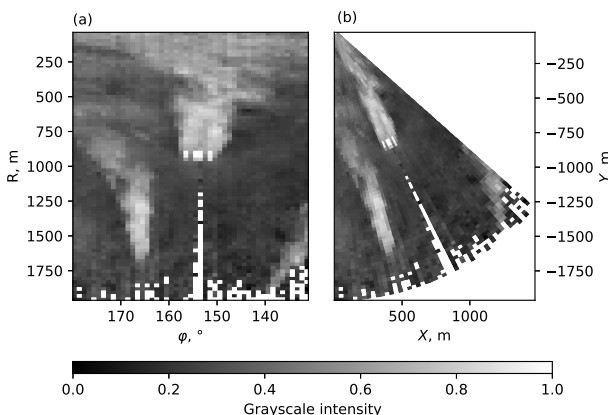

**Figure 2.** An example lidar scan #497 taken on September 24th 2016 19:18:20 UTC+0 at reference wind speed 7.4 ms$^{-1}$ and reference wind direction 151.14°. The original data are presented in (a) the polar coordinates $R, \varphi$ as stored in the matrix and (b) the Cartesian coordinates $X, Y$.

is scanned near the hub height at approximately 97 m. The farther wind turbines AV10 and AV11 are scanned above the top of
110 the blade tip at 158 m and 188 m, respectively.

The lidar measurements partially cover the day of September 24, 2016 and capture a variety of wake-wake interactions. The consecutive lidar scans are separated by approximately 45 s – the time required for the lidar to finish one scan. The data set contains 600 lidar scans, which are split into 24 subsets of 25 scans. Each subset contains the first $20-22$ minutes of each hour. For the simplicity of presentation and referring, we number the lidar scans from 1 to 600.

The ATS algorithm accepts the input data as a grayscale image. The wind speed data of each lidar scan are normalized by scaling to the range of [0, 1] to imitate the grayscale intensity as

$$I = \frac{U_{\max} - U}{U_{\max} - U_{\min}}, \tag{1}$$

where $U$ is the wind speed measured at a point, $U_{\min}$ and $U_{\max}$ are minimum and maximum wind speeds registered in a particular lidar scan. For the lowest wind speed $U = U_{\min}$ (potential wake points), $I = 1$ denotes the points with the highest
intensity. Similarly, the highest wind speed for $U = U_{\max}$ (free-flow points), $I = 0$ indicates the points with the lowest intensity.

The wake identification is performed on the data stored in a polar coordinate matrix (Fig. 2a). For a better presentation, the resulting data are plotted in the Cartesian coordinates as a scanned sector (Fig. 2b).

## 2.2 Large-eddy simulation

We also perform a large-eddy simulation to demonstrate and verify the performance of the ATS method and compare it against
the Gaussian wake identification and characterization method described further in Sect. 4.3. We use the PALM LES code with a built-in actuator disc with rotation (ADR) wind turbine model (Maronga et al., 2020). The results produced with the model were shown to capture the reduction of the wake deficit with the downstream distance at the rate similar to the encountered for

wind turbines (Vollmer et al., 2015, 2017; Doubrawa et al., 2020). The wake recovery aspect is particularly important to test the ATS method performance in the far wake. The currently used polynomial kernel also allows fitting the Gaussian function to compare it with the ATS method.

The domain contains $2304 \times 576 \times 192$ points and has the horizontal grid spacing of $4\,\mathrm{m}$. The vertical spacing below $600\,\mathrm{m}$ is also $4\,\mathrm{m}$. Above $600\,\mathrm{m}$, the vertical spacing is stretched with a factor of 1.08, capped at maximum $8\,\mathrm{m}$ grid cell height. The roughness length of $z_0 = 0.0005\,\mathrm{m}$ corresponds to the calm sea surface. The Coriolis forcing is enabled for the latitutde of $54°$, and the wind speed components are set to $u = 10.5\,\mathrm{ms}^{-1}$ and $v = -2.6\,\mathrm{m}^{-1}$ so that the flow rotation is compensated and the flow is aligned with the $x$-axis resulting in the horizontal speed of $10\,\mathrm{ms}^{-1}$ at the hub height. The surface temperature is $277\,\mathrm{K}$ and increases by $1\,\mathrm{K}$ per $100\,\mathrm{m}$. Neither heat flux nor surface heating are activated. During the simulation the turbulence intensity reaches $6.6\%$.

The reference NREL 5MW wind turbine has a hub height of $102\,\mathrm{m}$ and a diameter of $D_r = 126\,\mathrm{m}$ and is placed in the center of the domain so that the wake length can reach up to $20D_r$.

The LES is used solely to generate idealized wake data. No direct comparison to the lidar data is performed.

## 3 Lidar data pre-processing and classification

### 3.1 Data quality

Working with the current data set, we encountered two types of noise affecting the quality of the wake identification through thresholding: small wind speed fluctuations not directly caused by the wake and high wind speed values appearing due to a measurement error.

The measurement errors are primarily caused by the difference between wind direction and lidar orientation. The lidar measures radial velocity, which can be represented through three directional wind speed components $u$, $v$, and $w$, and the information on the line of sight of the lidar beam, given by the azimuth $\phi$ and elevation angle $\theta$:

$$U = u\sin\phi\cos\theta + v\cos\phi\cos\theta + w\sin\theta \tag{2}$$

When the wind blows along the lidar's line of sight, the measured radial velocity is essentially the horizontal wind speed. With the wind direction differing from the line of sight, the radial velocity deviates from the actual wind speed magnitude. In the case of crosswind – the wind direction is close to a perpendicular to the line of sight – the radial velocity tends to zero and does not represent the actual wind speed. The measurements taken during the crosswind event are more prone to errors compared to other wind directions.

When plotted against the reference wind direction, the reference wind speed and mean radial wind speed of a lidar scan show strong discrepancy for a range of wind directions (Fig. 3). With the lidar scanning in the range of $131.5-179.5°$, the crosswind effects can be expected for the wind directions of $221.5-269.5°$. As shown in Fig. 3, the crosswind effects already appear for the wind direction above $210°$. The scans taken near crosswind direction show a large number of non-physical wind speed values reaching $100-1000\,\mathrm{ms}^{-1}$. We refer further to these scans as 'corrupted'.

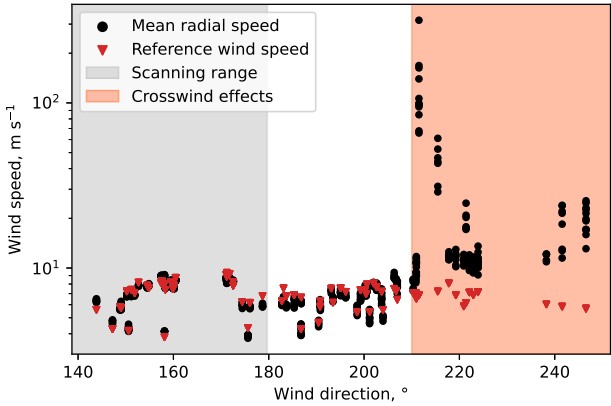

**Figure 3.** Comparison of the mean radial wind speed and the reference wind direction in the data set.

Occasionally, we also observe weaker spikes in the radial wind speed, most of which are localized at the position of a wind turbine AV10, implying a measurement error due to the lidar beam reflection from rotating blades. The reference and mean radial wind speeds remain in the good agreement for the wind directions below $210°$ despite containing spikes in the wind speed data. Nevertheless, the outliers cause an intensity skew when the wind speed data are normalized to the range of [0, 1] (Fig. 4). The intensity distribution peak moves to the right, with the left side containing occasional low bumps caused by the spikes (Fig. 4d).

In the example, the middle scan (Fig. 4b) has a wind speed spike of $15\,\mathrm{ms}^{-1}$, while the reference wind speed reaches $5.8\,\mathrm{ms}^{-1}$. The radial wind speed magnitude measured in the spike region stays below $7\,\mathrm{ms}^{-1}$. The lidar scan after normalization is less contrast compared to the adjacent lidar scans.

To preserve the uniformity between consequent lidar scans of the same subset, we perform despiking – detection and removal of the spikes. The spikes are detected based on the wind speed value and the difference with the adjacent points. We delete all values higher than $30\,\mathrm{ms}^{-1}$ and check the remaining data for the local maximums. An empirically chosen wind speed difference of $7\,\mathrm{ms}^{-1}$ proved to be enough to designate a local maximum as a spike. When a spike consists of a single or double point, it is replaced by a NaN value and the resulting gap is filled by interpolation to retain the continuous wind field. Three or more adjacent points designated as spike are considered a noise cluster; in such cases, gap filling after removal is not performed.

Since the lidar is oriented towards the closest wind turbine, a string of missing values – a wind turbine 'shadow' – is always present in the lidar scans regardless of the wind direction. The shadow rarely crosses wind turbine wakes and does not noticeably affect the performance of the wake detection methods. Hence we do not perform a gap filling to remove the shadow in addition to the despiking.

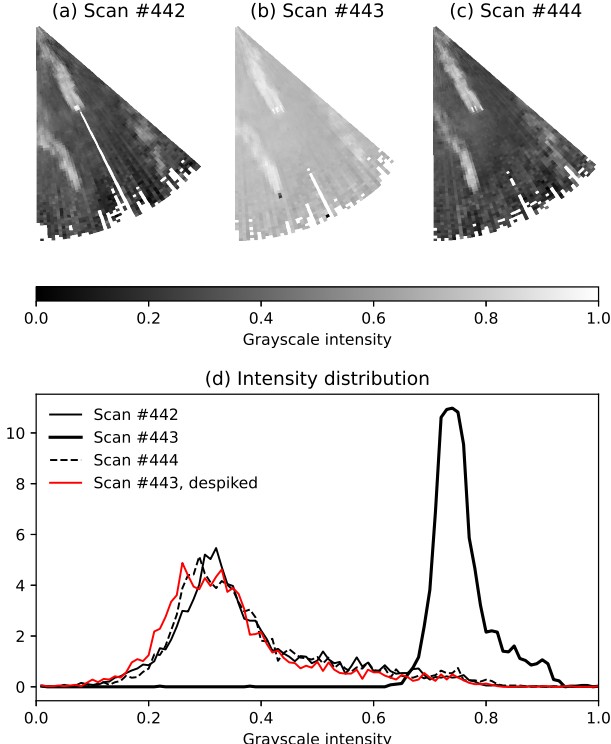

**Figure 4.** Consecutive lidar scans from the bimodal subset. No outliers are present in scans #442 and #444, while scan #443 has wind speed spikes near the wind turbine position and in the far range; subplot (d) shows the intensity distribution for the same scans.

## 3.2 Information entropy and data classification

We introduce entropy criteria as an alternative to using reference wind speed and direction for quality control. The entropy application ranges from finding a threshold (Pun, 1981) to object classification in an image (e.g., satellite map segmentation by Long and Singh (2013)). Here, we calculate it primarily for the diagnostic purposes and data classification into subsets.

The information entropy is a measure of noise in the data. It can be calculated for the whole data set as well as across the rows or columns of a rectangular matrix containing 2D data. We apply Shannon entropy $S$ (Shannon, 1948) as follows:

$$S = -\sum_{i=1}^{n} P(x_i) log_2 P(x_i), \tag{3}$$

where $P(x_i)$ is the probability density function (PDF) of the variable $x_i$ (here – intensity) to occur in the data. If the entropy tends to zero, it indicates uniform data. A high entropy value implies disturbances in the lidar scan due to wakes or noise.

To analyze lidar scan features, we calculate entropy for the partial data instead of the whole scan. We select wind speed values either in the radial or azimuthal direction and calculate a PDF of this sample to pass it to the entropy function. An example is presented in Fig. 5. The top and the left parts of the example scan in polar coordinates do not contain wakes,

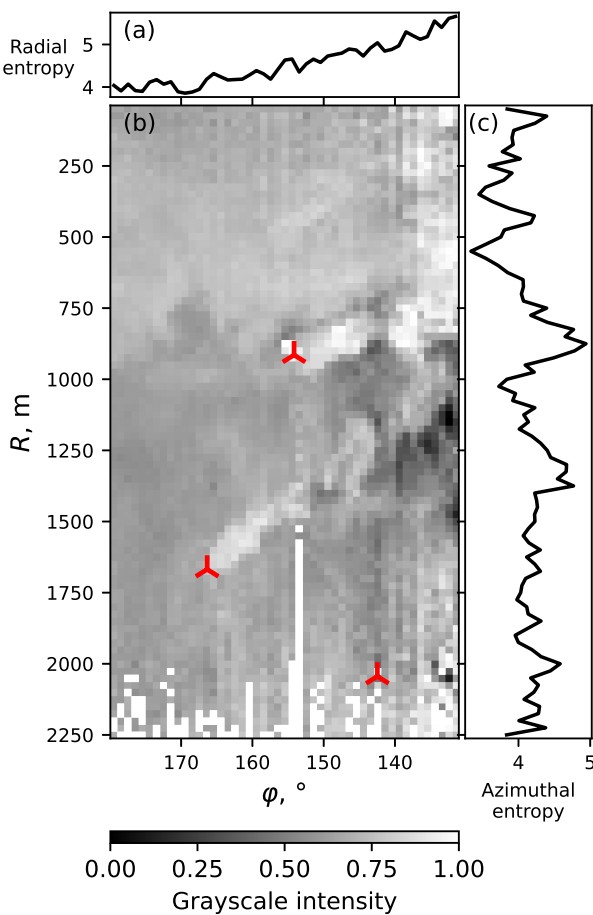

**Figure 5.** The entropy calculated in the (a) radial and (c) azimuthal directions of (b) the lidar scan #61. Reference wind speed is $7.19\,\mathrm{ms}^{-1}$ and reference wind direction is $203.68°$

hence the entropy calculated for the respective rows and columns is lower than for the wake regions. The entropy calculated in the radial direction (Fig. 5a) is higher for columns crossing both wakes instead of one due to higher disturbance rate. An additional entropy increase near the azimuth of $130-140°$ can be explained by high noise at the lidar scan border. The entropy calculated in the azimuthal direction (Fig. 5c) shows a peak for the AV7 wake. The AV10 far wake produces less prominent peak indicating a wake spread along longer distance and not aligned with the azimuthal direction.

We calculate the entropy in radial and azimuthal directions for all lidar scans before preprocessing. Combined into two plots, the entropies present an overview of the data set (Fig. 6). The respective wind turbine positions are marked on the right axis. The lower color bar limit is adjusted for the better presentation of the features contained in non-corrupted scans. For the scans with low noise, the entropy values fall into the range of $4-5$ both in the azimuthal and radial direction. The entropy calculated in the azimuthal direction highlights several lidar scans with a substantial entropy decrease (Fig. 6a) – the value drops below

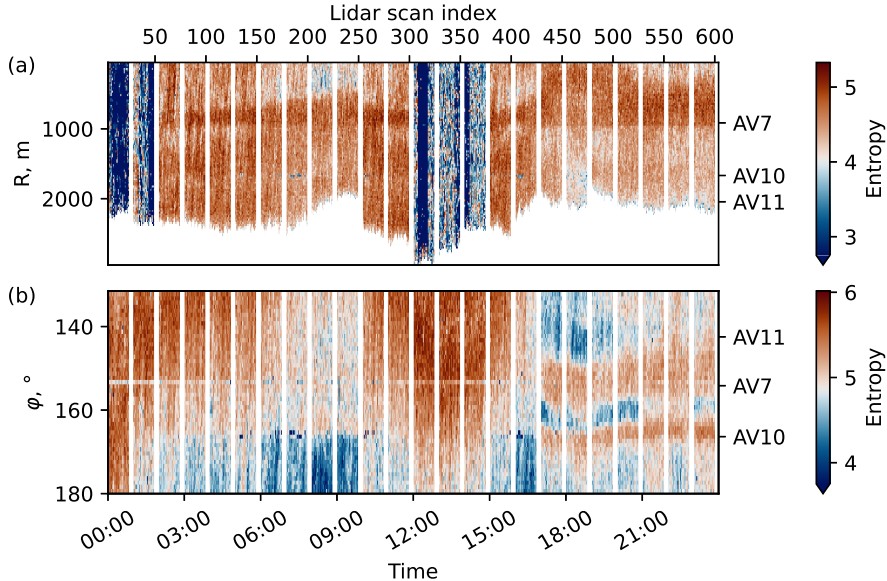

**Figure 6.** Entropy of the raw lidar data, all 600 lidar scans combined: (a) azimuthal entropy, (b) radial entropy.

two and tend to zero. The same scans are also characterized by the measurements corrupted due to the crosswind effect. The spiked data in non-corrupted scans leads to a local entropy decrease, seen as occasional blue dots mostly at the location of AV10. Series of such points can be seen for scans #176-200 and #401-410.

Non-corrupted subsets show similar entropy distribution in the azimuthal direction (Fig. 6a). A wake from the wind turbine AV7 can be seen as an increase of entropy near the turbine's location. A weaker increase of entropy can be also seen for AV10, for example, in scans #51−175.

    The entropy calculated in the radial direction is distributed uniformly for the corrupted subsets (Fig. 6b), but otherwise does not have as strong difference to non-corrupted data, as the entropy in the azimuthal direction (Fig. 6a). Part of non-corrupted

scans (#51−300 and #376−425) show a gradient-like pattern caused by the absence of wakes in 170−180° sector (low entropy) and wakes and border noise in 130−140° sector (high entropy). The pattern is weaker for scans #176−250, where the border noise is absent and wakes are aligned along the line connecting wind turbines, thus disturbing a smaller area of a lidar scan. The scans #426−600 combined demonstrate a horizontal stripe pattern, caused by the wind blowing towards the lidar. Wakes forming across the scanned azimuths cause the entropy increase in the radial direction matching the positions of AV7 and

AV10, as marked on the graphs.

    The low entropy criterion agrees well with the crosswind criterion on which scans are likely to contain a high amount of corrupted data. In general, the scans with a high corruption rate can be identified based on the percentage of the data points exceeding a specific wind speed limit. Since the reference wind speed does not exceed $10\,\mathrm{ms}^{-1}$, we consider the wind speeds

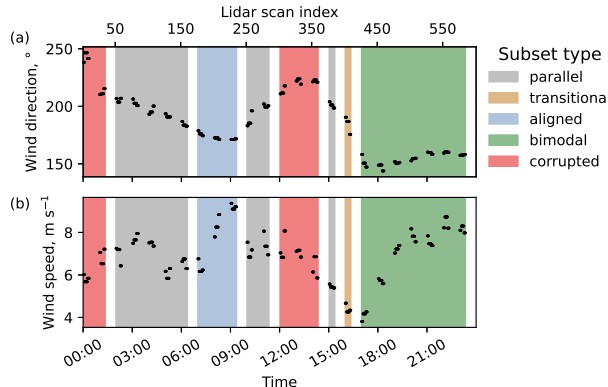

**Figure 7.** Classification of the subsets and overview of the reference wind direction (a), and wind speed (b).

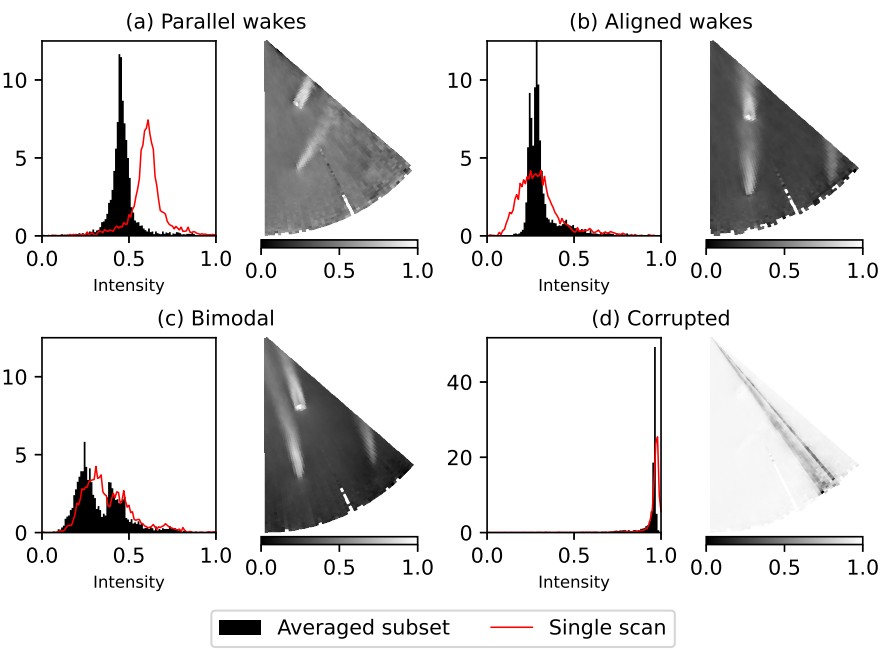

**Figure 8.** Sample averaged subsets and intensity histograms corresponding to the averaged subset and a single scan within the subset. The wind speed data are normalized to imitate the grayscale intensity. Despiking and removal of non-physical wind speeds are not performed to preserve the characteristics before preprocessing.

above $30\,\mathrm{ms}^{-1}$ to be a likely measurement error. The corrupted scans consistently have at least 1% of points exceeding this limit. The percentage drops to $0-0.05\%$ for the rest scans and corresponds to the occasional spikes.

The number of corrupted scans is 125, i.e., about 1/5 of the total number of scans. Classification of the remaining valid scans requires either a priori knowledge of the reference wind direction (which may be unavailable if we work with image data)

or visual evaluation of the wake features (which may be complicated for a large data set). Entropy criteria can simplify the classification by presenting a condensed overview of the data set. Using the entropy and intensity histograms, we classify the subsets into the following groups:

1. **Parallel wakes subset**, Fig. 8a: The wakes do not interact with each other. Some noise may occur at the lidar scan's border due to the wind direction approaching the value where the crosswind effects start. Since the wakes propagate towards this border and add to the disturbance, the entropy calculated in the radial direction shows a consistent increase near the azimuth of $131°$. The entropy calculated in the azimuthal direction shows a strong increase near the location of AV7 due to the wake and a weaker disturbance caused by AV10. The intensity histogram of an averaged subset tends to be more symmetrical than in other subsets and has a peak close to the intensity of 0.5. The intensity histogram of a single scan has a peak deviating from the center depending on the amount of noise. Parallel wakes are the most common case for this data set.

2. **Aligned wakes subset**, Fig. 8b: The wind blows along the line connecting wind turbines AV7 and AV10 so that the former is subjected to a wake. The entropy patterns are generally similar to the parallel wakes subset, except that a footprint of the AV10 wake is no longer visible for the entropy calculated in the azimuthal direction. The wind direction is closer to the scanned azimuths range and measurements have less noise compared to the parallel subset, hence the scans show slightly lower entropy. Compared to the parallel wakes subset, the histogram peak is shifted to the left. The histogram peak may split into two small peaks located close to each other when the wakes are not perfectly aligned.

3. **Transitional subset**. The wind direction changes, so both parallel and aligned wakes can be observed in the subset. This behavior is observed for a single subset containing scans #401−425. The transition to slightly lower entropy can be seen for the entropy calculated in the radial direction at azimuths 130-150° (Fig. 6b).

4. **Bimodal subset**, Fig. 8c: The wind blows along the lidar beam. Two long wakes are formed behind the wind turbines and merge in the lidar near range. Since the near range is scanned at a high resolution (Fig. 2), the far wake is represented by a larger percentage of points compared to the other subsets. Consequently, the intensity histogram approaches a bimodal distribution, which is especially prominent for the averaged subset. The larger peak represents the free flow, while the smaller peak corresponds to the far wakes of AV7 and AV10. The two peaks may merge resulting in one flat peak. The scans have little noise; the increase of entropy, especially in the radial direction, highlights the presence of the wakes.

5. **Corrupted subset**, Fig. 8d: The lidar scan is characterized by the amount of non-physical measurements (wind speed higher than $30\,\mathrm{m/s}$) exceeding 1% of the lidar scan points. While the valid measurements still take the largest share of a single scan, they are now considered 'low' wind speeds in a comparison to the maximum value. Due to the normalization (Eq. 1) that converts low values into light pixels, the histogram tends to the far right side, forming a sharp peak in intensity values between 0.9 and 1.0. The entropy in the azimuthal direction is lower than in other subsets and approaches zero, while the entropy in the radial direction tends to be more uniform than in non-corrupted scans and does not react to the presence of a wake.

**Table 1.** Overview of the lidar data subsets.

| Data type | Subset | Scans | WSPD, m s$^{-1}$ | WDIR, $^\circ$ | Entropy | % of data |
|---|---|---|---|---|---|---|
| Parallel wakes | 3 | 51−75 | 6.99 | 205.3 | 5.11 | |
| | 4 | 76−100 | 7.71 | 202.7 | 5.29 | |
| | 5 | 101−125 | 7.48 | 196.0 | 5.38 | |
| | 6 | 126−150 | 6.05 | 191.4 | 5.35 | |
| | 7 | 151−175 | 6.58 | 184.0 | 5.01 | 33.3 |
| | 11 | 251−275 | 7.10 | 187.8 | 5.12 | |
| | 12 | 276−300 | 7.41 | 200.2 | 5.37 | |
| | 16 | 376−400 | 5.45 | 200.9 | 5.02 | |
| Transitional | 17 | 401−425 | 4.38 | 184.5 | 4.76 | 4.2 |
| Aligned wakes | 8 | 176−200 | 6.32 | 176.2 | 4.69 | |
| | 9 | 201−225 | 8.30 | 172.2 | 5.28 | 12.5 |
| | 10 | 226−250 | 9.19 | 171.3 | 5.30 | |
| Bimodal | 17 | 426−450 | 4.11 | 151.5 | 5.44 | |
| | 18 | 451−475 | 5.71 | 147.3 | 5.31 | |
| | 19 | 476−500 | 7.22 | 150.9 | 5.72 | |
| | 20 | 501−525 | 7.83 | 154.2 | 5.67 | 29.2 |
| | 21 | 526−550 | 7.52 | 159.4 | 5.64 | |
| | 22 | 551−575 | 8.46 | 160.1 | 5.72 | |
| | 23 | 576−600 | 8.16 | 157.7 | 5.70 | |
| Corrupted | 1 | 1−25 | 5.80 | 243.3 | 1.54 | |
| | 2 | 26−50 | 6.85 | 212.1 | 2.31 | |
| | 13 | 301−325 | 7.27 | 213.4 | 1.53 | 20.8 |
| | 14 | 326−350 | 7.06 | 222.2 | 2.64 | |
| | 15 | 351−375 | 6.41 | 222.0 | 2.52 | |

The overview of the subsets and reference values is presented in Fig. 7 and Table 1 containing wind speed, wind direction and entropy averaged over each subset. A sample histogram averaged for a typical subset from each group is shown in Fig. 8 together with a single scan histogram from the same subset.

## 4 Methodology

Wake detection includes two stages (Quon et al. (2020)): wake identification – a separation of the wake from the free flow, and wake characterization – further analysis of the identified wake. We focus on the wake identification methods, particularly, an identification method using thresholding and also provide an algorithm for the wake characterization through centerline detection from the thresholded data.

### 4.1 Wake identification using automatic threshold detection

The thresholding methods split an image into background (in our case – free flow) and foreground (wake). Despite lidar data having a considerable amount of disturbances in the free flow, the wind speed distribution in a lidar scan tends to have one peak, either sharp or flattened (Fig. 8). The wake points take a small share of the lidar scan stored in polar coordinate matrix, while the remaining points belong to the free-flow – i.e., the most prominent peak contains free-flow points. The exception is the bimodal subset, where the far wakes characterized by high number of points (Fig. 2). As the result, the histogram of a scan from the bimodal subset may have two peaks depending on the intensity of the far wakes. To make our wake identification method universal, we build it upon threshold detection from a single histogram peak. The specifics of the wake identification in the bimodal case are further described in Section 6.4 and in the Appendix.

A single peak limits the applicability of the common thresholding methods that search for the local minimum of a bimodal histogram (Otsu, 1979). The lidar scan structure has similarities with ocean surface images: a background with small disturbances and bright whitecaps. Bakhoday-Paskyabi et al. (2016) described three methods of an automated threshold detection for the whitecaps. We choose an Adaptive Thresholding Segmentation (ATS) method identified to be fast and reliable by the original study. The basic principles of the ATS method are introduced here on a test example of an instantaneous LES wake.

Figures 9a and 9b show the wind speed field of an instantaneous LES wake and the same data normalized to the range of [0, 1]. A threshold $T$ is an intensity value in the range [0, 1] that separates free flow and wake points. After the threshold is applied to the normalized wind field, a binary matrix $WP$ is constructed from $I$ as follows:

$$WP(i,j) = \begin{cases} 0: & I(i,j) \leq T \text{ – free-flow point,} \\ 1: & I(i,j) > T \text{ – wake point.} \end{cases} \qquad (4)$$

The intensity threshold can be converted back to the radial velocity threshold $U_{th}$ by reverting the normalization expression Eq. 1 as

$$U_{th} = U_{\max}(1 - T) + U_{\min}T. \qquad (5)$$

The normalized wind speed data are represented as an intensity histogram (Fig. 9c). Let $H(x)$ for $x \in [0, 1]$ be the cumulative distribution function (CDF) of the intensity data. Then $H'(x)$ and $H''(x)$ are its first and second derivatives, respectively. With respect to the definition of intensity $I$ in Eq. (1), the wake points are located in the histogram's tail, while the free-flow points form a peak on the left side. The transition region where the peak tends to the tail is a good choice to search for a suitable

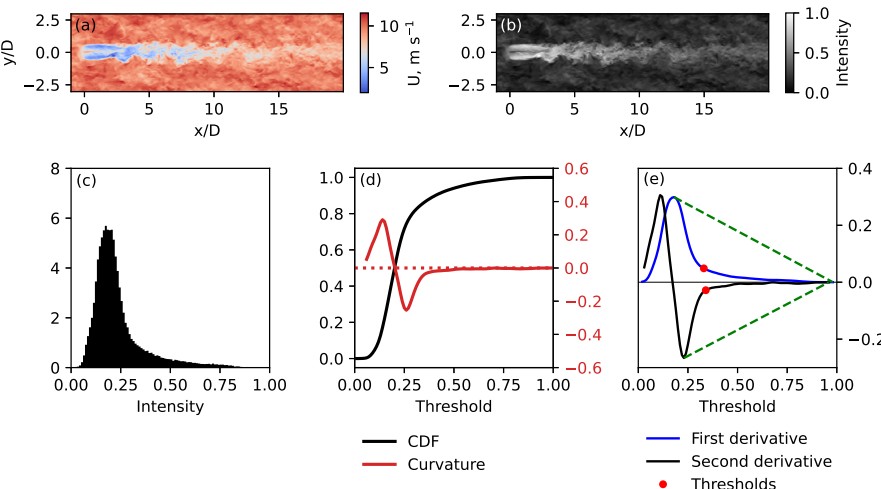

**Figure 9.** Sample LES wake and threshold detection. (a) Original instantaneous flow; (b) same flow normalized to the range of [0, 1]; (c) the intensity histogram of the normalized data; (d) CDF of the normalized data and CDF plot curvature; and (e) first and second derivatives of the CDF and the estimated thresholds.

threshold. We detect the threshold at the point where the CDF slope is close to constant, i.e., the curvature $C(x)$ approaches zero.

$$C(x) = H''(x)[1 + H'(x)^2]^{-3/2} \tag{6}$$

The curvature graph tail (Fig. 9d) may fluctuate and complicate the detection of the zero curvature. Instead, we look at the first and second derivatives $H'(x)$ and $H''(x)$ separately. The threshold value $T_2$ is selected as an inflection point at the right side of the second derivative graph (Fig. 9e). A similar point in the first derivative graph $T_1$ is used as a control value. We select the threshold as an average value between first and second derivative inflection points to smooth the threshold detection outcome. If the points initially laid close to each other, the averaged threshold $T = (T_1 + T_2)/2$ would not deviate too far from $T_2$. If the difference between $T_1$ and $T_2$ is high, the smoothing prevents the threshold from being too strict and leaving weak wakes undetected.

In the case of the lidar data, the derivative plots have strong oscillations. Therefore, we fit a polynomial function on the range between intensity $I_k$, corresponding to the most prominent local extremum and maximum intensity $I_{max} = 1$. We fit a function $F(k) = a_1 + a_2/k^5$, since the corresponding fit returned low root-mean-square error (RMSE) while not altering the inflection point location significantly.

After the threshold is found, we apply it to the data as described in Eq. (4) and obtain a binary matrix $WP$ that represents thresholded data. Each matrix point corresponds to an image pixel. Because of the wake irregularity, especially in the lidar scan, the method usually detects several clusters of high-intensity points. Any cluster may be a part of a wake as well as falsely detected noise. We do not yet distinguish between wake and noise and refer to all detected clusters as 'wake shapes'. Due to

---

**Procedure 1** Automatic threshold detection

---

**Input:** U(r, $\phi$) {raw lidar data}

    despike U(r, $\phi$)

**Input:** $k \in [0,1]$ step 0.01

    $I \leftarrow (U_{\max} - U)/(U_{\max} - U_{\min})$

    $H \leftarrow f(I,k)$ {get the intensity histogram}

    $H_1 \leftarrow \partial H/\partial k$

    smooth $H_1$ with moving average $n = 4$

    $H_2 \leftarrow \partial^2 H/\partial k^2$

    smooth $H_2$ with moving average $n = 4$

    normalize $H_1$ and $H_2$

    $P_1 \leftarrow$ fit $F(k) = a_1 + a_2/k^5$ on $[k(\max H_1), 1]$

    $P_2 \leftarrow$ fit $F(k) = a_1 + a_2/k^5$ on $[k(\min H_2), 1]$

    $T_1 \leftarrow P_1$ inflection

    $T_2 \leftarrow P_2$ inflection

**Output:** $T \leftarrow (T_1 + T_2)/2$

    {threshold data as $WP(r,\phi)$}

    **if** $I(r,\phi) < T$ **then**

        $WP(r,\phi) \leftarrow 0$ {free-flow point}

    **else**

        $WP(r,\phi) \leftarrow 1$ {wake point}

    **end if**

---

the code implementation, the detected points belong to the same shape as long as the constituting points are adjacent in the matrix $WP$. The shapes touching only by the corners are considered to be separate shapes.

## 4.2 Wake characterization from the data thresholded by the ATS method

For the wake characterization, we detect the centerline of a wake shape. The centerline search method starts with extracting a contour of a wake shape; the further algorithm is based upon the geometrical properties. It should be noted that the centerline search algorithm does not strictly depend on the ATS method and can be used as a stand-alone algorithm that requires thresholded data as an input.

To start the centerline search method, we require a procedure to determine which shapes were correctly identified as a wake. The ATS method searches for the high-intensity points corresponding to the highest wake deficit. Containing the highest wind speed decrease, the near-wake region perfectly satisfies this condition. Therefore, it can be expected that the near wake will be one of the largest continuous shapes among the detected and will contain a wind turbine within or near it. The borderline

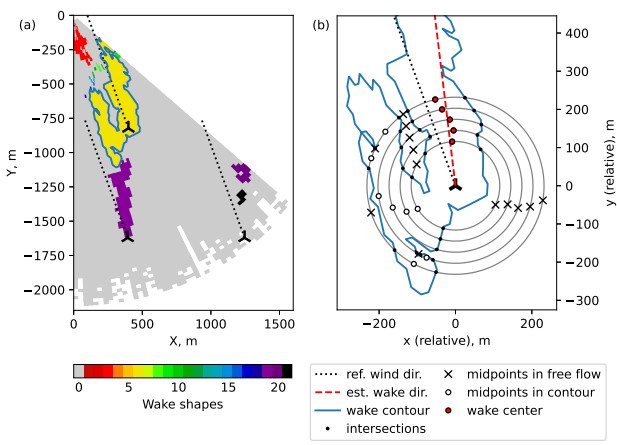

**Figure 10.** An example wake centerline detection in an ambiguous case.

contour of such shape is extracted for further analysis. The wake centerline is then defined as a centerline of the extracted contour.

Assuming the outline of the shape as wake boundaries, we estimate the wake centerline using the following algorithm:

1. The algorithm starts by drawing a circle of radius $1D$ around the wind turbine and marks points where the circle crosses the borders of the wake shape. If the circle appears to lie within the wake shape completely, the initial radius is increased until intersections are found.

2. The midpoint of the arc inside the wake contour indicates the wake direction and is stored as the centerline midpoint.

3. The circle diameter is increased by a pre-defined step, e.g., $0.1D$, and the steps $1-2$ are repeated until the end of the wake shape is reached.

This short algorithm works as it is for an ideal case of a smooth wake contour and known wind direction matching the wake direction. However, the circular lines may cross the irregular wake contour several times. Considering the near wake to be wide and continuous, we expect the centerline point to lie within the wake shape. We also assume that the wake does 330 not turn gradually further downstream. Therefore, the segment between the last known and unknown midpoint should turn by a relatively small angle compared to the previous segment. The wind or wake direction is advantageous to distinguish wake shapes from noise, as it allows narrowing the search by disregarding shapes detected in the upwind direction as false detection.

The currently used centerline search algorithm 2 provided further includes these and several other rules for selection of a wake point when an ambiguity is present. Fig. 10 shows an example of resolved ambiguity based on the wind direction. If the 335 wind direction was not available, an estimated wake direction could be used instead with the same outcome.

Generally, this centerline-search method does not require a priori knowledge of the wind direction. However, it may be difficult to resolve the ambiguity on the first step, if the wind direction is unknown. For example, the aligned wakes subset

**Procedure 2** Wake centerline detection
___

$N(r,\phi) \leftarrow$ label $WP(r,\phi)$ {enumerate detected shapes}

**Input:** $(x_w, y_w)$ or $(r_w, \phi_w)$ {wind turbine coordinates}

  **if** $WP(r_w, \phi_w) = 1$ **then**

    $n \leftarrow WP(r_w, \phi_w)$ {select a wake shape containing the wind turbine}

  **else**

    find $n \in N$ and $R$ {select the largest detected shape near the wind turbine location}

  **end if**

  $L \leftarrow$ boundary contour of the wake shape $n$

  **for** $d = 1$ to $15D$ step $0.1D$ **do**

    $C \leftarrow$ contour of a circle with radius $r$ centered at $(x_w, y_w)$

    $p^i \leftarrow$ intersect $L$ and $C$

    calculate midpoints on the arc between $(p^i, p^{i-1})$

    $N_p \leftarrow$ number of midpoints inside the wake shape $n$

    **if** $N_p == 0$ **then**

      $(x_c^r, y_c^r) \leftarrow$ NaN {the circle does not cross the wake contour}

    **else if** $N_p == 1$ **then**

      $(x_c^r, y_c^r) \leftarrow$ midpoint $p_1, p_2$ {centerline point is the only midpoint inside a wake}

    **else**

      {ambiguous centerline point, limit the search}

      $\alpha \leftarrow$ deviation from the know wind or wake direction for each valid midpoint

      $(x_c^r, y_c^r) \leftarrow \min(\alpha^{(x_c^r, y_c^r)})$

    **end if**

  **end for**

**Output:** $X_c, Y_c$
___

(Fig. 8b) and, to a certain extent, also the bimodal subset (Fig. 8c) introduce ambiguity in a wake direction for the downstream wind turbine AV7. A circle drawn around AV7 may cross the detected wake in at least four points. The algorithm will in turn identify downstream and upstream points as potential centerline points. To continue the search, the algorithm has to select only one direction. In the absence of the reference wind direction, the ambiguity can be resolved by approximating the wake direction first.

The procedure to approximate the wake direction runs similarly to the centerline search, with few alterations. First, the step is increased, but the algorithm is run for a shorter length until $4D$ downstream, so only the most well resolved part is processed. All midpoints laying inside the wake contour are accepted, since there is yet no way to make a distinction between them. A linear function $y(x) = ax$ is then fit to the identified midpoints. If the coefficient of determination is negative ($R^2 < 0$), the fit is too inaccurate, and the procedure is repeated for another wind turbine.

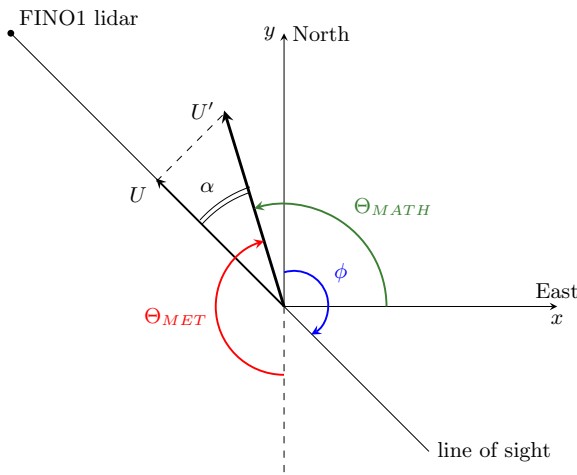

**Figure 11.** The relation between radial wind speed $U$, actual wind speed $U'$, azimuth angle $\phi$ and meteorological and mathematical wind directions $\Theta_{MET}$ and $\Theta_{MATH}$.

The intercept value $a$ of the best fit is the arctangent of the mathematical wind direction (Fig. 11). The approximated meteorological wind direction is then

$$\Phi_{MET} = \frac{3}{2}\pi - \Phi_{MATH} = \frac{3}{2}\pi - \arctan a \tag{7}$$

The approximated wake direction may strongly deviate from the actual wake direction, so it is only used to resolve ambiguity. The actual wind direction is estimated from the full centerline. We convert the coordinates of the centerline points for the AV7 and AV10 wakes to the Cartesian system and subtract the respective wind turbine positions to get a set of the relative centerline coordinates. We assume a centered data set and add a point $(0, 0)$ corresponding to the relative wind turbine position. The wake from the wind turbine AV11 is prominent only for the bimodal subset and is too short and easily confused with the noise in the other subsets. We do not consider this wake in our analysis due to the little information it can provide compared to the other two wakes. The composed data set is fitted with the linear regression, and the fitted line indicates the estimated wake direction.

### 4.3   Wake identification and characterization using the Gaussian method

The wake deficit distribution is similar to the Gaussian distribution in the far wake (Ainslie, 1988) and often shows a double Gaussian peak in the near wake (Magnusson, 1999). The similarity to the Gaussian distribution makes a base for a widely used method to detect wake boundaries and centerline (Vollmer et al., 2016; Krishnamurthy et al., 2017). The method requires the data in a two-dimensional horizontal plane, which makes it versatile and practical to use for the wake identification and characterization.

Due to the lidar elevation angle, AV10 is scanned near the top tip and does not show a double wake. The scan resolution near AV7 is not always sufficient to resolve a pronounced double wake. Therefore, we fit the wake deficit distribution with a

single Gaussian function:

$$F(y) = A \exp\left(-\frac{(y-\mu)^2}{2\sigma^2}\right), \tag{8}$$

where the amplitude $A$, mean value $\mu$, and standard deviation $\sigma$ are the parameters to fit; the variable $y$ is a coordinate on a line perpendicular to the wind direction. The fitting starts from $1D$ to avoid uncertainties caused by a weak double wake observed for AV7. We attempt fitting for the wake deficit profiles up to $15D$ downstream distance covering the length of most wakes in the lidar data set.

For a wake deficit distribution, the fitted Gaussian function $F(y)$ reaches its maximum at $y = \mu$, i.e., the estimated mean $\mu$ gives the wake center position. The wake boundaries are defined through the mean value $\mu$ and the standard deviation $\sigma$ as $\mu \pm 2\ln 2\sigma$ so that the velocity deficit at the wake boundaries is 5% of the velocity deficit at the wake center (Aitken et al., 2014).

The Gaussian function is fitted to the wake deficit of $\Delta U(y) = 1 - U/U_0$, thus a knowledge of the free-flow wind speed $U_0$ is also required. Since the background flow is rather non-uniform in the lidar scans, we probe the velocity at each cross-section at $1.6D$ from the rotor axis (Krishnamurthy et al., 2017). The LES data use the wind speed at the hub height as the free-flow wind speed.

We run the Gaussian method in an automatic mode. The method should be applied to the data extracted along the straight-line perpendicular to a pre-defined search direction. The algorithm thus requires knowledge of the wind direction before the fitting. The algorithm is also dependent on the accuracy of the wind direction measurements and the similarity between reference wind and actual wake direction. During our analysis, we observed an offset of about $5°$ between the directions, which caused fitting errors for otherwise clear wake. To reduce the influence of a possible discrepancy between wind and wake direction, we recalculate the search direction every five points by fitting the linear function to the previously found center points.

The wake deficit profiles extracted for fitting have a width of $2.5D$, except for the bimodal subset. There, the profile width is decreased to $1.75D$ after reaching the downstream distance of $6D$. The correction is active only for the scans after #500 where the AV10 far wake and the AV7 wake come close enough, but do not yet merge completely and allow separation. If the Gaussian function is fit to a wider profile there, the fitting would mistake higher deficit in the AV7 near wake for the center of the AV10 wake. Reducing the extracted wake profile width improves the centerline detection in the AV10 far wake and delays the first occurrence of this error, although does not always prevent it.

## 4.4 Deficit-based wake identification

In addition to the Gaussian fit, we apply a threshold based on the wake deficit criterion. The method assumes that a point belongs to the wake if the wind speed there is less or equal to 95% of the free-flow wind speed (España et al., 2011); here – the reference wind speed.

The lidar measures radial velocity $U$ (Eq. (2)). If the wind direction differs from the scanned azimuths, the reference wind speed measured by a cup anemometer noticeably deviates from the free-flow radial velocity. Normally, a lidar retrieval procedure should be performed to reconstruct the actual wind field. Since we are only interested in the wind speed values, but not

the local flow direction, we apply a simple expression to re-project the radial velocity and take the magnitude of the calculated
wind speed.

$$U'_{r,\phi} = \frac{U_{r,\phi}}{\cos\alpha} \tag{9}$$

where $U_{r,\phi}$ is the measured radial velocity at the beam range $r$ and azimuth $\phi$; $U'_{r,\phi}$ is the estimated magnitude of the real velocity, and $\alpha$ is the angle between the radial and actual wind speed vectors (Fig. 11). Equation (9) assumes that the flow moves in the reference wind direction at each scanned point regardless of the wake influence and other flow disturbances.

The angle $\alpha$ is calculated as the difference between reference wind direction $\Theta_{MET}$, given according to the meteorological convention, and the azimuth $\phi$ (Fig. 11). I.e., Eq. (9) changes to

$$U'_{r,\phi} = \frac{U_{r,\phi}}{\cos\alpha} = \frac{U_{r,\phi}}{\cos(\Theta_{MET} - \phi)} \tag{10}$$

Since the normalization (Eq. (1)) is not performed, the deficit-based method does not necessarily require despiking – all high-value outliers would be assigned to the background flow by the threshold condition. However, the method requires additional
410 information on the free-flow, such as the wind speed and direction, to perform the simple retrieval.

The threshold is applied to the wind speed field recalculated with Eq. (10) instead of the original radial velocity field used for the ATS method. Therefore the direct comparison of the thresholds is complicated. Instead, we compare the thresholded images and evaluate the detection accuracy against the manual wake identification.

### 4.5 Manual wake identification and characterization

We perform a manual segmentation to select an optimal threshold for each lidar scan and use it as a 'true' identification. The manual threshold is defined in a way to represent the minimum threshold required to identify a wake shape suitable for the automatic centerline detection as described in Sect. 4.2. The comparison against manual wake identification then would show, whether the ATS method is capable to automatize the threshold selection and improve its flexibility compared to the deficit-based thresholding.

Since the available scans represent different wake-wake interactions, the criteria for a reasonable threshold vary over the subsets. In order to reduce human error, we use the following qualitative criteria:

1. The shape of the wake should be distinguishable well enough not to be misinterpreted as noise.

2. The noise should be reduced near wind turbines AV7 and AV10 but is allowed near AV11 since its wake has low importance in this study.

3. The identified wakes from AV7 and AV10 should not merge to ease the centerline detection.

We also perform a manual centerline detection. A centerline is drawn over the lidar scan as a line or series of points. For further comparison with other wake characterization methods, it is converted to the Cartesian coordinates using a plot digitizer. Unlike the manual threshold detection, the manual wake characterization is more prone to errors, especially in the

far-wake region, where the wake becomes less distinguishable from the free flow. Due to ambiguity and complexity of the

manual centerline detection, we select only few lidar scans to demonstrate the methods' performance in the parallel, aligned and bimodal subset.

For brevity, the wake identification and characterization methods are further referred to as listed in Table 2.

**Table 2.** Summary of the wake detection methods

| Name | Main characteristics |
| --- | --- |
| Manual | **Input data**: Radial velocity field.<br>**Identification**: Threshold value based on the visual evaluation.<br>**Characterization**: Digitized centerline drawn over the lidar scan.<br>**Automation**: No.<br>**Flexibility**: Yes. |
| Deficit-based | **Input data**: Retrieved velocity field, Eq. (9), reference wind speed.<br>**Identification**: Threshold value based on the wake deficit compared to the free flow.<br>**Characterization**: Not performed.<br>**Automation**: Yes.<br>**Flexibility**: No. |
| Gaussian | **Input data**: Radial velocity field, wind direction, wind turbine locations.<br>**Identification**: Gaussian function fitted to the wake profile.<br>**Characterization**: Performed simultaneously with the wake identification.<br>**Automation**: Yes.<br>**Flexibility**: Partial. |
| ATS | **Input data**: Radial velocity field, wind direction (optional), wind turbine locations.<br>**Identification**: Threshold value from the intensity histogram.<br>**Characterization**: Midpoints of the concentric arcs crossing the wake contour.<br>**Automation**: Yes.<br>**Flexibility**: Partial. |

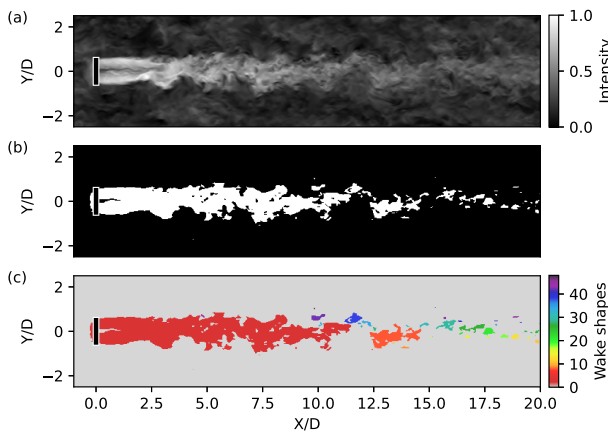

**Figure 12.** Wake and centerline identification for a sample instantaneous LES wake: (a) normalized flow field, same as Fig. 9b; (b) thresholded flow field; and (c) wake shapes color-coded to show connectivity.

## 5 Proof of concept: Wake identification and characterization from the LES data

In this section, we demonstrate the performance of the ATS method in application to the LES data and compare the result to

435 the Gaussian method.

An instantaneous LES wake reveals complex spatial features to be detected, although its intensity histogram remains rather smooth (Fig. 9b). The ATS method detects a continuous structure in the near wake and the beginning of the far wake, while the wake at $x/D > 10$ is represented as series of small disconnected structures (Fig. 12b-c). The ATS method does not capture the wake expansion, but only a trail of the low wind speed areas.

Since the ATS method extracts the outer contour of a shape, small holes inside the detected wake are automatically filled and do not affect the intersection-based centerline search (Fig. 13a). The current algorithm processes only the first continuous wake shape. Extending the centerline downstream requires a procedure to identify which of the small detected shapes actually belong to the far wake and the connection order. The former problem is more relevant for a lidar scan, which has less uniform background flow compared to the LES data.

Figure 13c compares the wake centerline and edges detected by the Gaussian and ATS methods. Both methods perform well in the range of $1 < x/D < 10$ and show good agreement on the same distance (Fig. 13c). Downstream ($x/D > 10$), the wake becomes weaker as it recovers to the free flow. If the wake deficit function becomes too flat to fit accurately, the fitting result may place the wake center incorrectly or overestimate the standard deviation and, consequently, the wake width. The ATS method detects only disconnected structures in the far wake. Nevertheless, those structures primarily lie within the wake edges

detected by the Gaussian method. The Gaussian centerline also passes through the centers of the ATS-detected structures. A good agreement between methods can be explained by the fact that the ATS method searches for regions of high intensity, i.e.,

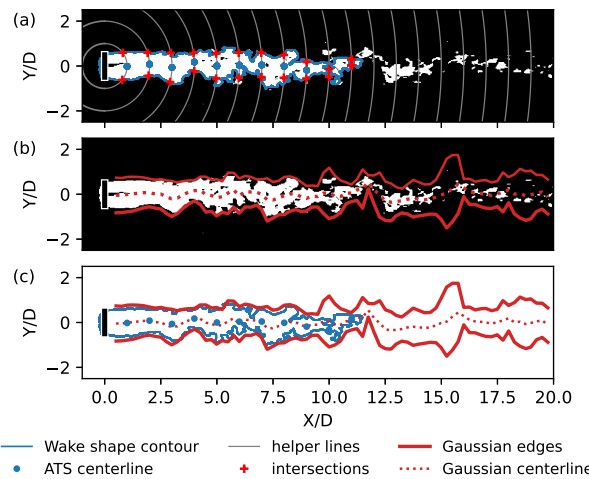

**Figure 13.** Sample wake identification and characterization using idealized LES data. (a) Thresholded data overlaid with the contour of the wake shape; (b) thresholded data overlaid with the wake boundaries and centerline detected by the Gaussian method; (c) ATS and Gaussian wake detection results, overlaid.

low wind speed. At the same time, the Gaussian method approximates a wake center at the point of high wake deficit, which also corresponds to low wind speed.

Overall, the Gaussian and ATS centerline search method show complimentary flaws. The Gaussian method may estimate
the wake center correctly on a weak wake profile, but return a large standard deviation leading to an overestimation of the wake width. The Gaussian method does not always interpret strong wake meandering correctly and mistakes a wake turn for a wide wake. On the contrary, the ATS method is capable to discern a complex wake shape, but has problems with the centerline detection if the wake shape is too irregular due to wake merging or mixing with noise.

## 6 Results

For the lidar data, we perform an extensive comparison to the manual wake identification and characterization and evaluate the accuracy of the ATS method. We further compare the performance of the ATS and Gaussian methods and discuss the application of the ATS method in the centerline detection. We show both ensemble statistics and demonstrate the methods performance on sample scans showing each of the most represented non-corrupted subsets: parallel, aligned wakes and bimodal.

### 6.1 Comparison of the ATS wake identification against the manual identification and deficit-based thresholding

We construct a confusion matrix to assess the performance of the methods for a single lidar scan. The $2\times2$ confusion matrix describes the comparison of the automatic thresholding methods (deficit-based or ATS, see Table 2) against the manual method and contains the following outcomes:

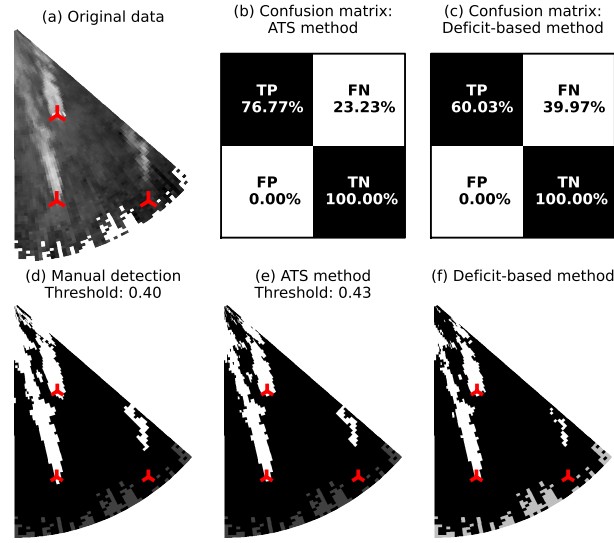

**Figure 14.** Scan #599 (bimodal subset), wake identification. (a) The original data in the Cartesian coordinates; (b-c) confusion matrices for the ATS and deficit-based methods; (d) manual threshold selected in a way to separate the two wakes; and (e-f) thresholds estimated by the ATS and deficit-based methods.

- True Positive (TP) – the point is detected as a wake point by both manual and automatic identification.

- True Negative (TN) – the point is detected as a free-flow point by both manual and automatic identification.

– False Positive (FP) – the point is detected as a wake point by the automatic method but is a free-flow point in the manual identification.

- False Negative (FN) – the point is detected as a free-flow point by the automatic method but is a wake point in the manual identification.

If the automatic identification is accurate with respect to the manual identification, TP and TN values tend to 100%, while
FP and FN are close to zero.

The bimodal subset can be considered the most convenient for the manual threshold segmentation. It utilizes the strict criterion for the manual threshold that the wake shapes should not merge (Fig. 10d). In the example, the ATS method sets the threshold higher compared to the manual identification (Fig. 10e). Hence the far-wake area is slightly reduced. The deficit-based method (Fig. 10f) produces a similar result.

The aligned wakes subset utilizes the same manual threshold criteria for the wake splitting as the bimodal subset (Fig. 15), although the condition may be harder to fulfill. For some lidar scans, the far wake from the turbine AV10 and the near wake from AV7 cannot be separated, unless the threshold is increased so that the far wake is not identified (Fig. 15d). In this case, detecting a general shape of the wake takes the priority. The manual threshold is then more subjective than that of the bimodal

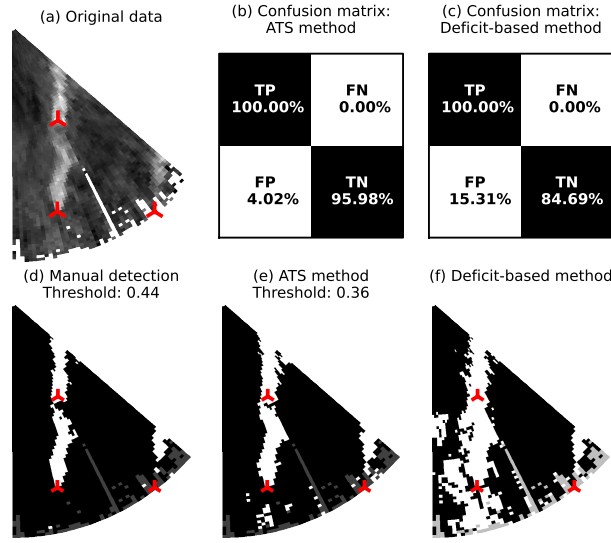

**Figure 15.** Scan #221 (aligned wakes subset), wake identification. (a) The original data in the Cartesian coordinates; (b-c) confusion matrices for the ATS and deficit-based methods; (d) manual threshold selected in a way to separate the two wakes; and (e-f) thresholds estimated by the ATS and deficit-based methods.

subset. The deficit-based method underestimates the threshold more significantly than in the bimodal case and produces larger percentage of false positives than the ATS method (Fig. 15f).

The parallel wakes subset is the most challenging, both for the manual identification and the automatic methods (Fig. 16). The wind direction in the subset is approaching $210°$, where the crosswind effects start (Fig. 3) and noise appears at the border of a lidar scan. Unlike the corrupted scans with a high amount of non-physical wind speed values, the region around the wind turbines AV7 and AV10 contains valid measurements and still allows performing wake identification with relative success. However, the wake identification accuracy declines due to the border noise, and only one wake can be extracted well enough to perform the analysis on the wake centerline and shape evolution. If the threshold is increased to distinguish wakes and noise, the wake from AV10 remains nearly undetected as can be seen from Fig. 16d. The ATS method returns a lower threshold that improves distinguishing the shape of the AV10 wake but falsely detects noise as a part of the AV7 wake (Fig. 16e). The deficit-based method estimates the threshold rather accurately, but may detect additional false positives near wind turbines (Fig. 16f).

We summarize the comparison of true negative and true positive detections in the box plots (Fig. 17) for the different subsets.

Due to the amount of noise, the parallel wakes subset is challenging for both methods. Nevertheless, the ATS method approaches manual identification rather effectively, while the deficit-based method leaves a decent amount of noise which may alter the identified wake shape (Fig. 16f).

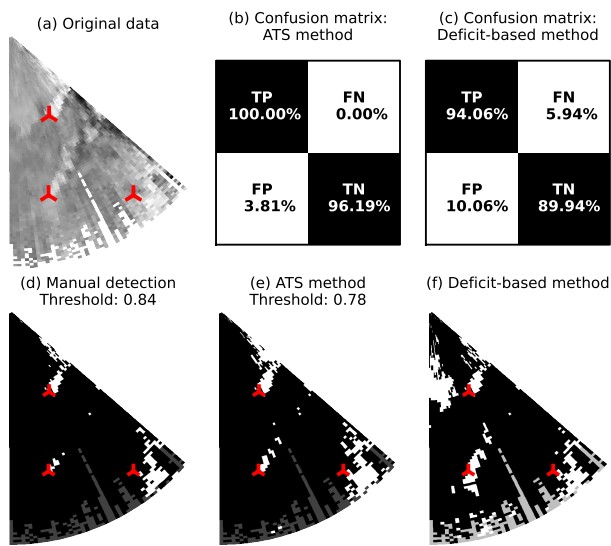

**Figure 16.** Scan #60 (parallel wakes subset), wake identification. (a) The original data in the Cartesian coordinates; (b-c) confusion matrices for the ATS and deficit-based methods; (d) manual threshold selected in a way to reduce noise but keep a general shape of the wakes; and (e-f) thresholds estimated by the ATS and deficit-based methods.

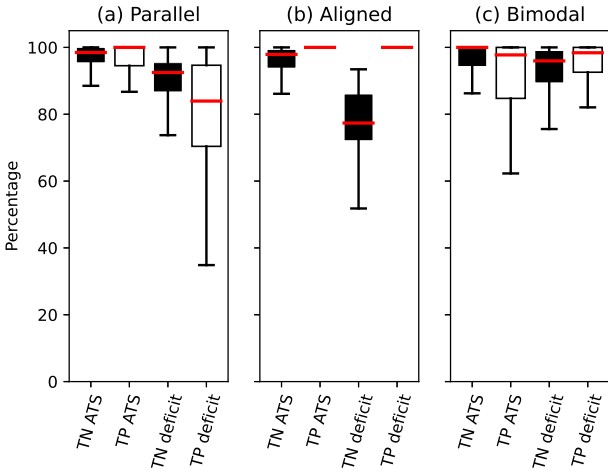

**Figure 17.** Ensemble statistics of true negative and true positive detections within the subsets.

Both methods score nearly 100% for the true positive detections in the aligned subset (Fig. 17b). The result is caused by the criterion for the manual threshold: separate two different wakes. The criterion is too strict for both automatic methods to achieve, therefore, they always underestimate the threshold. Still, the ATS method gets closer to the manual threshold, which is reflected in lower variation of true negative detections compared to the deficit-based threshold.

**Table 3.** Comparison of the thresholding methods' performance against the manual wake identification

| Data type | Subset | Scans | Manual-deficit, % | | | | Manual-ATS, % | | | |
|---|---|---|---|---|---|---|---|---|---|---|
| | | | TP | FN | FP | TN | TP | FN | FP | TN |
| Parallel wakes | 3 | 51−75 | 55 | 45 | 5 | 95 | 80 | 20 | 2 | 98 |
| | 4 | 76−100 | 69 | 31 | 4 | 96 | 97 | 3 | 4 | 96 |
| | 5 | 101−125 | 76 | 24 | 9 | 91 | 91 | 9 | 4 | 96 |
| | 6 | 126−150 | 85 | 15 | 9 | 91 | 95 | 5 | 4 | 96 |
| | 7 | 151−175 | 96 | 4 | 23 | 77 | 98 | 2 | 3 | 97 |
| | 11 | 251−275 | 95 | 5 | 22 | 78 | 99 | 1 | 4 | 96 |
| | 12 | 276−300 | 71 | 29 | 7 | 93 | 93 | 7 | 2 | 98 |
| | 16 | 376−400 | 80 | 20 | 6 | 94 | 96 | 4 | 3 | 97 |
| Transitional | 17 | 401−425 | 93 | 7 | 19 | 81 | 87 | 13 | 0 | 100 |
| Aligned wakes | 8 | 176−200 | 99 | 1 | 28 | 72 | 98 | 2 | 1 | 99 |
| | 9 | 201−225 | 100 | 0 | 13 | 87 | 100 | 0 | 10 | 90 |
| | 10 | 226−250 | 100 | 0 | 23 | 77 | 98 | 2 | 3 | 97 |
| Bimodal | 17 | 426−450 | 88 | 12 | 2 | 98 | 82 | 18 | 0 | 100 |
| | 18 | 451−475 | 83 | 17 | 1 | 99 | 89 | 11 | 0 | 100 |
| | 19 | 476−500 | 97 | 3 | 5 | 95 | 96 | 4 | 4 | 96 |
| | 20 | 501−525 | 97 | 3 | 7 | 93 | 85 | 15 | 2 | 98 |
| | 21 | 526−550 | 99 | 1 | 15 | 85 | 90 | 10 | 5 | 95 |
| | 22 | 551−575 | 100 | 0 | 20 | 80 | 90 | 10 | 8 | 92 |
| | 23 | 576−600 | 85 | 15 | 2 | 98 | 94 | 6 | 4 | 96 |

The deficit-based and ATS wake identifications behave rather similar for the bimodal subset (Fig. 17c) with respect to the manual wake identification. The variations in the bimodal subset are primarily caused by the wakes forming in lidar near range, which is scanned at higher resolution than the rest of a scan. I.e., any small threshold change affects more points at the wake edges than it would for the parallel or aligned subsets and results in stronger fluctuations in TP-FN values.

To reduce the influence of ambiguity of the manual detection, we construct a confusion matrix for each subset of 25 consecutive lidar scans instead of single scans. The corrupted scans are excluded from the comparison, since high noise prevented the manual detection for most of the scans. Table 3 summarizes the detection outcomes for each subset. The ATS and deficit-based method perform comparably in terms of true positives in the aligned and bimodal subsets. However, the amount of false positives for the deficit-based method indicates a high probability of identifying noise as a wake. Additionally, the percentage of false positives strongly fluctuates within the same type of the subset making the fixed threshold method unreliable.

While the amount of true positives for the ATS method may drop to 80% for a complex subset, the amount of true negatives consistently stays near 95% – the background flow is mostly detected correctly regardless of the subset type, which is an improvement compared to the deficit-based method. Compared to the manual detection, the ATS method does not always separate wake and noise correctly, particularly for the parallel wakes subset (Fig. 16) and thus requires additional filtering. For the aligned and bimodal subsets, the ATS method is capable to detect the general wake shape rather similar to the manual detection.

It should be noted that the deficit-based wake identification requires a free-flow wind speed to define the threshold and an additional preprocessing of a lidar scan – a correction based on the wind direction or a more complex lidar retrieval method. The ATS method runs solely on the lidar data and does not require information besides what is already contained in a lidar scan.

## 6.2 Comparison of the wake characterization using Gaussian and ATS methods

We perform the wake characterization by searching for the wake centerline from the thresholded image produced with the ATS method as described in Sect. 4.2 or by applying the Gaussian method as described in Sect. 4.3. First, we provide a comparison of selected scans against the manual wake characterization from the lidar scan image as described in Sect. 4.5. The found centerlines are compared by fitting the regression lines to the relative coordinates, so that each local coordinate system is centered at a selected wind turbine.

The parallel wakes subset (Fig. 18) contains a short but pronounced wake from the wind turbine AV7 and a long weaker wake from the wind turbine AV10. Since the AV10 wake is frequently detected as series of small disconnected structures, the current ATS method detects the centerline only for the first continuous shape, which rarely extends beyond the near-wake region. The manual and Gaussian wake characterization can be carried further into the far-wake region, but become rather uncertain as the far wake recovers to the free flow or mixes with the border noise. Considering the problems that the border noise poses for the wake identification in less clean scans (Fig. 16), the characterization outcome can be improved by excluding the near-border sector of $1-2°$ width from the identification process.

The aligned wakes subset (Fig. 19) shows a distinctive feature: the wakes are aligned along the line connecting two wind turbines resulting into the merge of the AV10 far wake and the AV7 near wake. Additionally, the connecting line is parallel to the $Y$-axis in Cartesian coordinates, so the centerline tends to $X = $ const when the wakes are perfectly aligned. Hence, the coefficient of determination $R^2$ either approaches zero or becomes negative and does not indicate the quality of the regression fit.

The bimodal subset (Fig. 20) has the longest wakes in the data set. The wake identification in the far wake (i.e. lidar near range) is hindered by wake merging and the narrowness of the scanned area. For example, the ATS method may underestimate the threshold and detect merging wakes as a single shape. The ATS-based threshold can be adjusted to guarantee the wake splitting. The adjustment is performed automatically by increasing the threshold with an increment of 0.05 until the stopping criterion – the wind turbines belong (or are located near) to different wake shapes – is reached.

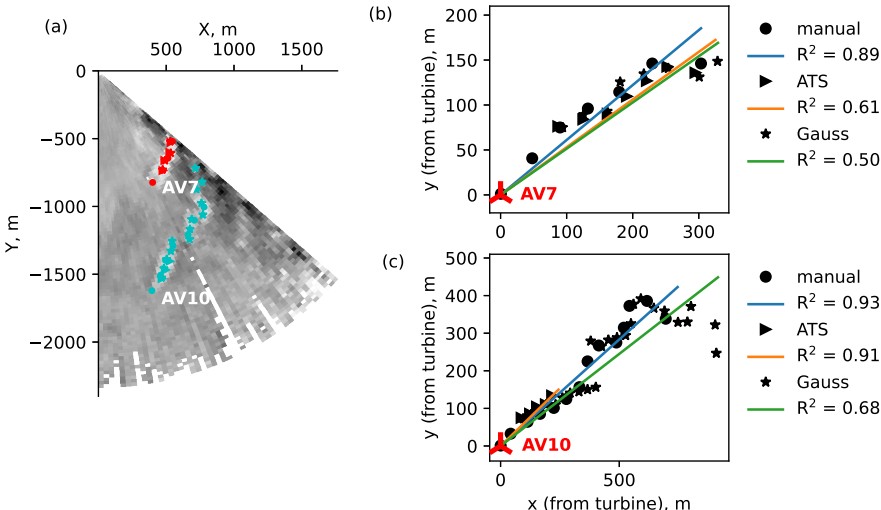

**Figure 18.** Scan #59 (parallel wakes subset). (a) An overview of the detected centerlines and regression fits for (b) AV7 and (c) AV10.

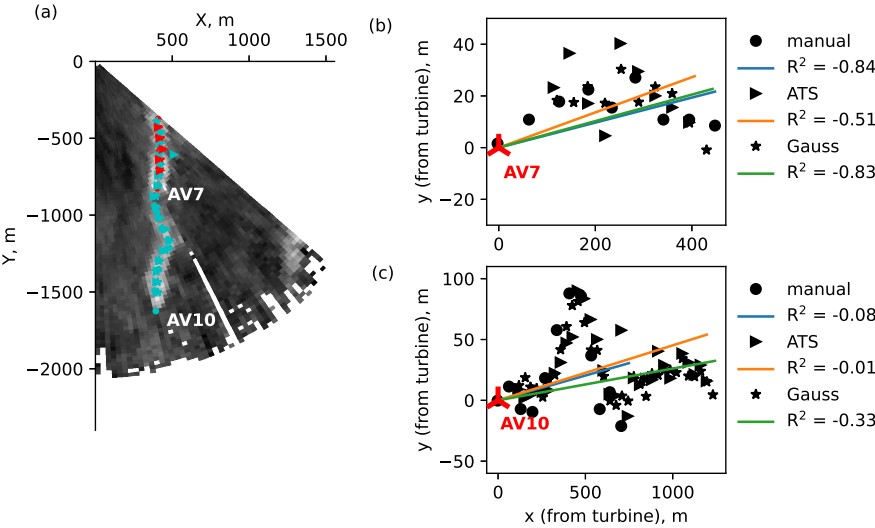

**Figure 19.** Scan #221 (aligned wakes subset). (a) An overview of the detected centerlines and regression fits for (b) AV7 and (c) AV10.

The merging wakes also affect the accuracy of the Gaussian method, as it may detect a wake center incorrectly because of high wake deficit in the neighboring wake. The characterization inaccuracy in the lidar near range is compensated by higher
overall number of data points available for fitting, compared to the other subsets.

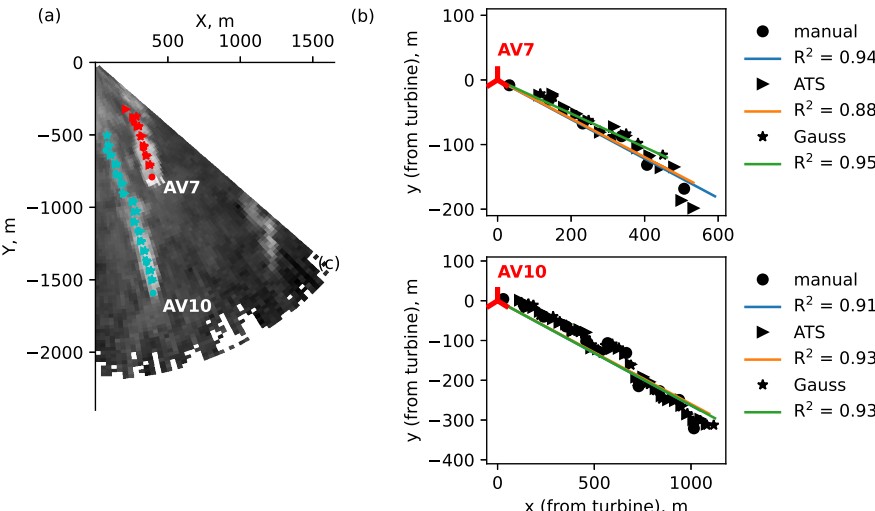

**Figure 20.** Scan #599 (bimodal subset). (a) An overview of the detected centerlines and regression fits for (b) AV7 and (c) AV10.

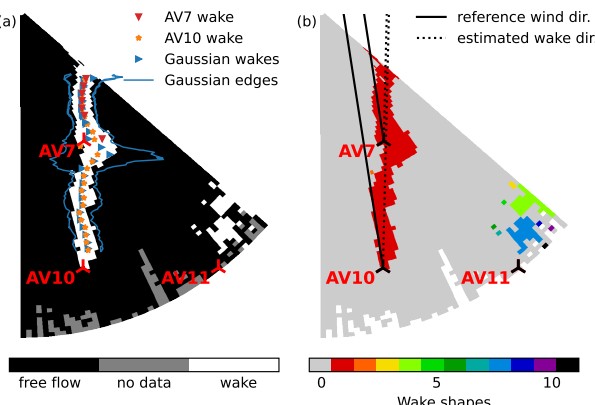

**Figure 21.** Scan #222 (aligned wakes subset) sample wake identification and characterization showing (a) comparison of the ATS and Gaussian methods, (b) wakes identified by the ATS method after the threshold is applied, and wind and wake direction.

Figure 21 shows an example of wake identification performed on a lidar scan from the aligned wakes subset. The subset is characterized by the wake merging near AV7. The formed structure proves to be challenging for a Gaussian method, as the centerline point and far-wake width for AV10 are estimated incorrectly.

The ATS method detects wakes as a single shape. Unlike the bimodal subset, the merging wakes in the aligned wake subset do not necessarily worsen the performance of the centerline detection method. The centerline is first detected for the AV10 wake, from which the wake direction can be estimated. Since the wakes are merged, the centerline detection for AV10 continues in the AV7 wake. The centerline search for AV7 starts at the corresponding turbine location and is performed in the direction of

**Table 4.** Rules for scoring Gaussian and ATS centerline search methods performance.

| Score | Gaussian method | ATS centerline search |
|---|---|---|
| 0 | The method failed to find the wake at all or less than 10% of the visible wake was identified. | |
| 0.5 | The centerline loosely matches the wake centerline, but the wake width is overestimated or undefined. | The wake shape is readable from the thresholded image, but the centerline is incomplete or erroneous. |
| 1 | The method had correctly identified at least 75% of the visible wake and its centerline. | |

the AV10 wake, thus excluding the merge region from the search. Thus the centerline of the AV7 wake gets detected twice if no stopping criterion (e.g., the AV10 centerline passes AV7 location) is activated. Both detected centerlines agree in the AV7 wake region and follow the Gaussian centerline rather well. Near-border wake centers of the AV7 wake deviate from the presumed centerline because border noise is erroneously attributed as a part of the wake.

When it comes to the comparison of wake characterization over the whole data set, the effect of weak wakes or merging on the Gaussian method performance complicates a direct comparison. Due to the errors, the Gaussian centerline cannot be taken as a 'true' value and requires verification on its own.

Instead, we perform a visual comparison of the Gaussian and ATS centerline search methods to score their success rate. The performance of both methods rather differs along the wake, therefore we evaluate the detection result on two segments: $l \leq 4D$ and $l > 4D$ from the wind turbine. The $l \leq 4D$ segment usually covers the most well-resolved part of the wake in non-corrupted scans, we attribute it as the near wake. The rest of the wake would be then referred as the far wake and characterized by lower wake deficit. Next, we score the success rate based on whether the method was able to identify both wake shape and centerline, failed on one of the tasks, or did not distinguish the wake at all (Table 4).

As mentioned for the LES wake identification and characterization (Sect. 5), the ATS and Gaussian methods are prone to errors in different aspects. A partial success for the Gaussian method would usually mean a centerline estimated with a large standard deviation, while a partial success for the ATS method would be the detection of the wake shape, but not the full centerline.

A summary for the data set excluding corrupted scans is presented in Fig. 22 by showing the counts for each outcome and their distribution between the subsets.

The near wakes are well resolved and show a high number of outcomes where both methods succeed. The partial detections are spread differently. The non-perfect outcomes for the AV7 near wake are spread rather equally (Fig. 22a). The increased

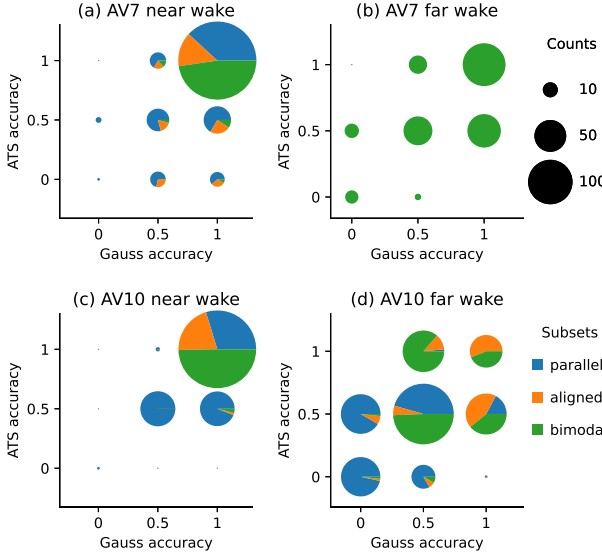

**Figure 22.** Overview of the Gaussian and ATS method performance on the wake detection and characterization.

error rate of the ATS method in the AV7 near wake is caused either by strong border noise (parallel subset) or strong upstream wake influence (aligned subset) – both distort the detected wake shape.

Due to the studied wind directions, the AV10 near wake is not subjected to the upstream turbine influence. The wake is very clear and poses problems mainly for the ATS method in the parallel subset, when it cannot be identified as a continuous shape. Hence, the ATS method under-performs and stops at the wake identification, while the Gaussian method can succeed in both aspects (Fig. 22c).

The comparison of AV7 far wake accuracy (Fig. 22b) is relevant only for the bimodal subset, where the corresponding wake reaches the required length. Detection outcomes for the AV7 far wake follow a pattern that resembles the other cases: very low counts of partial or full success when one of the methods fails, higher counts for partial and full success of both methods.

The exception from this pattern is the AV10 far wake (Fig. 22d). Both methods achieve partial success most often. The decreased success rate is primarily caused by the wake merging in bimodal and aligned subsets. When it comes to the parallel wake subset, both methods are likely to fail. The weak AV10 far wake limits efficiency of both methods: the threshold is not enough to separate the wake from the free flow and the fitting cannot be carried on to the nearly flat wake deficit function.

The low count of (0, 1) pairs throughout the comparison indicates that none of the methods outperforms another in any part of the wake. If one method fails, the other usually fails too or achieves only a partial success.

## 6.3 Wind and wake direction

The regression line fitted to the ATS-detected centerline also indicates the wake direction. A strong mismatch between reference wind direction and wake direction can be seen for most lidar scans from the data set (Fig. 21b).

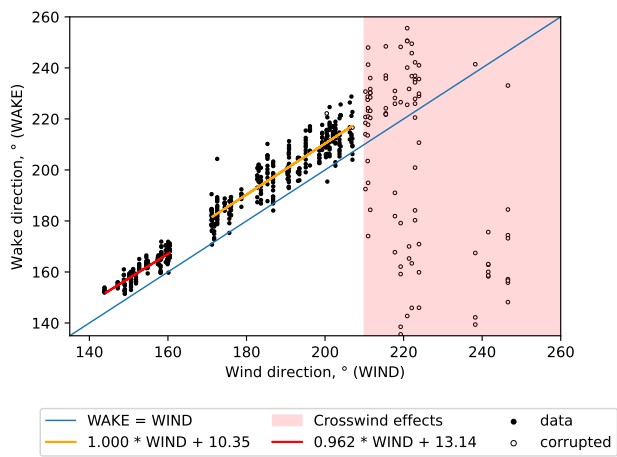

**Figure 23.** Comparison between reference wind direction (WIND) and estimated wake direction (WAKE).

Comparing the directions for the whole data set, we observe a clear trend for the wake direction deviating clockwise from the reference wind direction until the crosswind effects start at 210° (Fig. 23).

The valid points for the reference wind directions less than 210° group into two distinct clusters (Fig. 23). The leftmost cluster corresponds to the bimodal subset and lies within the range of wind directions of 140−170°. Another cluster contains the results for the aligned, transitional, and parallel wakes subsets and covers the range of wind directions of 170−210°. Fitting a linear regression to each group returns a similar slope but a different intercept value. Although the fitted line slope is not equal to one, the regression fit on the selected range shows a nearly constant offset between wind and wake direction with the bimodal subset having noticeably lower difference, than other subsets.

The vertical veer and clockwise rotation of the wake in the Northern hemisphere due to the Coriolis force are known effects causing wake rotation and were confirmed by observations and LES studies of wind farms (Magnusson and Smedman, 1994; Abkar and Porté-Agel, 2016; van der Laan and Sørensen, 2017). The wind turbine AV7, closest to the lidar, is scanned nearly at the hub height, while the farther wind turbines, AV10 and AV11, are scanned near the top-tip height (Fig. 1). Due to the elevation and vertical veer, the wind and wake direction discrepancy is the strongest for AV10 and AV11. Nevertheless, we also observe a deflection for the near wake of AV7, although the noticeable effects of the Coriolis force are usually recorded for the downwind distance of $6D$ or higher. The additional discrepancy can be explained by the yaw misalignment (Bromm et al., 2018), reference measurements uncertainty (Gaumond et al., 2014), and the lidar installation's imperfection. The wake direction variation for the bimodal subset (reference wind direction 140−160°) was possibly reduced because of the longer wakes and, consequently, more precise estimation of the wake direction. We do not have additional data to distinguish these factors and leave it for a future study.

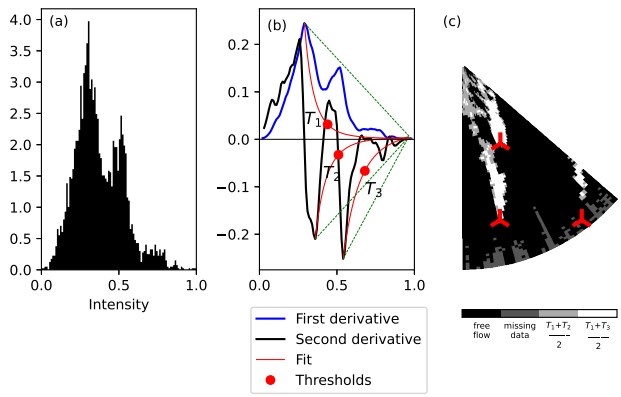

**Figure 24.** Scan #553 (bimodal subset). (a) Intensity histogram of the normalized data, (b) the ATS method search for the thresholds, (c) thresholded image.

The outliers showing strong differences between wind and wake direction highlight the lidar scans where the wake identification and characterization was hindered by noise or strong irregularity of the wake. The wind-wake direction plot can be used for diagnostic purposes to select the lidar scans that require additional processing prior to the wake identification.

## 6.4 Wake identification in the bimodal subset using the ATS method

Bimodal subsets often have a distinctive double peak in the intensity histogram (Fig. 24a). The highest histogram peak corresponds to the free flow. The second peak forms due to a long far wake from AV10 and subsequent merging of the two wakes.

     The double peak from the histogram translates into two local minimums in the second derivative graph (Fig. 24b). The occasions of two local maximums in the first derivative were rarer in the regarded data set. Applying the ATS method to both

second derivative minimums provides a unique opportunity to estimate two thresholds $T_2$ and $T_3$ in addition to the threshold $T_1$ from the first derivative. The final threshold values either separate the full wake from the free flow ($(T_1 + T_2)/2$) or extract only the most intense part of the wake ($(T_1 + T_3)/2$) (Fig. 24c). The splitting point falls approximately at the downstream distance of $4-5D$, marking a transition from the near to far wake.

     We ran the ATS method without subset-specific parameters meaning that it always estimated only one threshold for the wake

identification. During the threshold estimation (Sect. 4.1), the current algorithm selects the global maximum or minimum of the first and second derivatives, respectively. The free-flow histogram peak usually results in the global maximum of the first derivative in our data set and does not affect the performance of the ATS method. However, the local minimum values of the second derivative appear to be more sensitive to the intensity distribution. Relying on the global minimum may lead to selecting a stricter threshold $T_3$ (Fig.24b). A strict threshold does not detect most of the far wake, as shown in Fig.24c.

A less strict threshold $T_2$ could be chosen based on the proximity to $T_1$ as a control value. However, it would require an automatic check whether another local minimum can produce a valid threshold. The implementation posed a challenge, if the

current algorithm ran without subset-specific parameters, and produced erroneous threshold estimation for other scans. We refrained from using more complex approach in the bimodal subset for now. The current ATS method, therefore, overestimated the threshold and did not identify the full wake in about 8% of the bimodal cases.

## 7  Conclusions

We developed a set of methods to analyze lidar scans for wake identification and characterization. During the study, we focused on the procedures that would automatically process a large data set and primarily rely on the information contained in the lidar data or site characteristics such as lidar and wind turbine positions. To structure the analysis of the results, we split our data set into several subsets grouping the scans with similar characteristics. While the classification could be performed based on the wind direction or visual inspection, we introduced entropy as criterion to reflect the flow characteristics. When calculated in the azimuthal or radial direction, Shannon entropy is sensitive to the disturbances caused by wakes and allows scan classification if the wind direction is unknown. The entropy values also highlighted the lidar scans that were unsuitable for the analysis due to the high amount of non-physical measurements caused by the crosswind effects. The classification by entropy criteria introduced in the study was not yet used to apply scan-specific corrections during the thresholding.

An existing automatic thresholding method, the deficit-based method as referred in the study, thresholds the wind speed data at 95% of the free-flow wind speed and was initially suggested for more regular wind tunnel wakes. The reliance on the actual wind speed hinders the deficit-based method performance on the lidar data – a retrieval procedure should be applied to the measured radial velocity to reconstruct the wind field. Additionally, the fixed ratio of 95% does not regard the quality of a lidar scan. To overcome these disadvantages, we proposed an automatic thresholding method for wake identification, the ATS method, based on the method for whitecaps detection on the ocean surface. The method did not require the knowledge of the actual wind speed and could be applied to the radial velocity data. The preparatory step applied normalization through scaling data to the range of $[0, 1]$, thus requiring the removal of outliers during the lidar scan preprocessing.

The comparison to the manual thresholding showed that the ATS method generally performed better than the deficit-based method and on the par with the manual wake identification, which opened a possibility to use it when manual thresholding is infeasible.

We also described an automatic method for the wake centerline search from the thresholded data. The centerline search could run without wind direction provided by making a rough approximation of the wake direction. However, the current algorithm processes only the first continuous shape limiting the application to the wind fields with little noise obstructing the wake identification.

We compared the centerline found from the thresholded data to the Gaussian fit method. Although the Gaussian method performance on the lidar scans was not as good as on the LES data, the wake characterization in the near-wake region showed an agreement between the methods with respect to the manual centerline detection. At the same time, the accuracy of both Gaussian and ATS-based methods decreased in the far-wake region, especially for noisy data or in the case of wake-wake interaction. In the latter case, the ATS method often identified two wakes as a single shape, affecting the centerline search

algorithm. The algorithm performs better when the wake directly hits the downstream wind turbine – the merged wakes can be considered as one wake and have a common centerline. When the wakes are forming side by side and get close to each other, the threshold may need additional adjustment until the identified wake shape is split.

The results showed that automatic thresholding from the intensity histogram was viable for the wake identification not only for the LES but also lidar data. We see a potential to improve the wake characterization algorithm to detect the centerline of 675 the whole wake and plan to present it in future studies.

*Code and data availability.* The Python code for wake identification using ATS method, centerline detection and a sample lidar data set are available upon request.

*Video supplement.* The video https://doi.org/10.5446/54055 demonstrates wake identification results for all lidar scans in the data set. No post-processing is performed after running the ATS algorithm.

**Appendix A: Image data processing**

An image has several properties which may affect the algorithm performance compared to the use of the raw wind speed data:

1. Image resolution in dots-per-inch (dpi): the resolution of 72 dpi transforms an original data point into an image pixel as 1-to-1 approximately. Higher resolution increases the number of pixels per data point. Lower resolution merges several data points into one pixel.

2. Colormap: the ATS method relies on the image grayscale intensity as an input. A non-grayscale image can be desaturated, but the colormap of the original image then should be sequential rather than perceptually uniform or diverging. For the latter, the conversion to the grayscale gradually reduces the contrast between high and low values making the wake identification impossible. Additionally, several grayscale colormaps exist. Depending on the colormap, the intensity histogram of an image may shift to the left or right compared to the raw data. We observed this effect when the 'Greys' 690 colormap of the Python Matplotlib library was used. This colormap emphasizes light tones; as a result, the intensity histogram peak slightly shifts to the right, although the general shape of the peak is preserved (Fig. A1). The colormaps 'binary' or 'gray' from the same library return the result that follows the original data.

3. Image intensity: as processed by Python, the values are rounded up to second digits and some are assigned to different bins compared to the original data. The histogram and CDF have stronger oscillations than the raw data (Fig. A1) and 695 require smoothing before the application of the ATS method.

Running an automated threshold detection on the image raises another question: how much does the image resolution affect the identification accuracy compared to the raw data. We apply the ATS algorithm to raw and image data under different

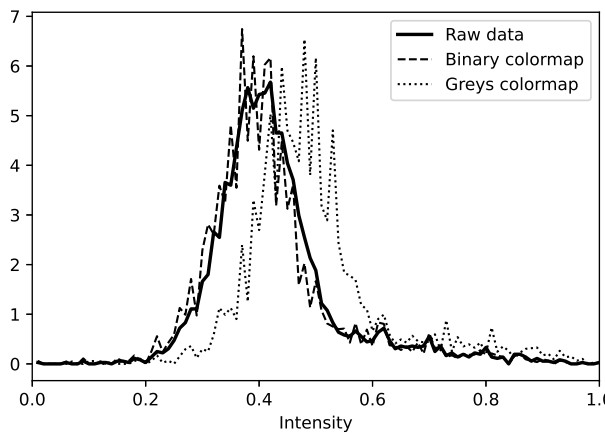

**Figure A1.** Comparison of the intensity distribution in the original (raw) data and image plotted using Python Matplotlib with different grayscale colormaps.

resolutions: 72, 150, and 300 dpi. We observe little influence from the image resolution, except for few LES cases, where the low resolution of 72 dpi affected the threshold detection. In those cases, the detected threshold is lower than in fine resolution cases, and, therefore, a larger shape is identified as a wake. The image resolution of 150 dpi and above agrees well with the wake identification from the raw data. The general shape of an image intensity histogram does not depend on the image resolution. The image resolution of 150 dpi or higher is recommended for use, although 72 dpi also produces good detection results.

In the case of lidar measurements, the wake identification from the image data can be performed in two ways: by plotting the original data either in polar or Cartesian coordinates. The wake identification from the polar coordinates image does not bear a notable difference from the raw data, apart from the aforementioned specifics of the image resolution and intensity. However, if the lidar data are plotted in the Cartesian coordinates as a scanned sector, the lidar close and far range get distorted, affecting the percentage of the area covered by the wakes and, consequently, the histogram shape.

The effect is most pronounced when the wind blows towards the lidar. As described in the subsets overview in Sect. **??**, this wind direction and wake behavior result in the bimodal intensity histogram. The leftmost, high peak, contains points from the free flow, while the second, low, peak accumulates points from the far wake. The second peak gets smoothed when the input data are changed from the normalized wind speed to the grayscale image plotted in Cartesian coordinates. As can be seen from the comparison (Fig. A2), the lower peak corresponds to the data in the lidar's close range. After the conversion to the Cartesian coordinates, the close range area shrinks significantly, while the free-flow area on the far lidar range enlarges. The transition between coordinate systems changes the balance between wake and free-flow pixels and virtually increases the share of the latter.

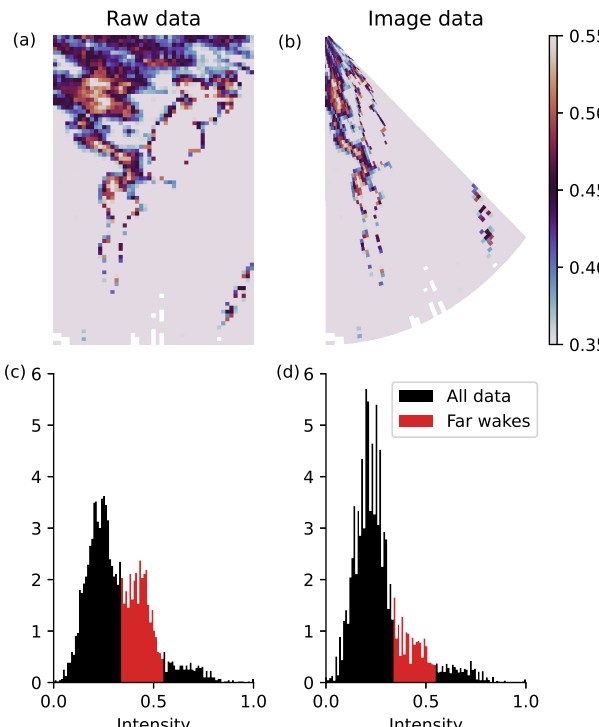

**Figure A2.** (a-b) Normalized wind speed data with far wakes highlighted and (c-d) corresponding grayscale intensity histograms of the lidar scan #551.

*Author contributions.* MK adapted the image processing algorithm developed by MBP to the wake identification, expanded the algorithm with the centerline characterization code and performed analysis of the wake identification and characterization. MBP provided further guidance on the image processing techniques. JR provided information on the lidar setup and explanation on the discrepancies between reference and lidar data. FGN consulted on the method practical application. All authors had contributed to the proofreading and correcting the article draft.

*Competing interests.* The authors declare that they have no conflict of interest.

*Acknowledgements.* The OBLEX-F1 field campaign has been performed under the Norwegian Centre for Offshore Wind Energy (NOR-COWE), funded by the Research Council of Norway (RCN) under project number 193821. The scanning Doppler wind lidar system (Leosphere WindCube 100S), used for this study, has been made available via the National Norwegian infrastructure project OBLO (Offshore Boundary Layer Observatory) also funded by RCN under project number 227777. The LES simulations for the study have been performed by using the high performance computer facilities of the Norwegian e-infrastructure Uninett Sigma2 (project number NS9506K).

The authors would like to thank Martin Flügge for providing the lidar data.

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
