# Peer review of "Development of an automatic thresholding method for wake meandering studies and its application on the data set from scanning wind lidar"

_Wind Energy Science, 2021_

## Referee Comment (RC1)

The work presents an interesting, new technique for wake identification and characterization from lidar scans using principles from image processing. The work presents promising results for wake characterization using a novel image thresholding method (applied to another use case in a separate field and used for the first time for wake characterization in this manuscript). Compared to existing velocity deficit thresholding techniques, this method exhibits a higher success rate for wake characterization. The paper is interesting and novel and considers the use of a LES simulation case study to explain the novel thresholding technique and compares multiple methods for wake characterization (Gaussian, velocity deficit threshold, the novel thresholding technique, manual inspection). However, these comparisons are hard to follow and the paper could benefit from a better organizational structure and improved figures. Ensemble statistics for successful wake characterization are only given when comparing the new thresholding technique and the velocity deficit thresholding technique previously used in the field. The paper could benefit from further clarification as to whether they are considering the far wake in their analysis and from displaying ensemble statistics comparing the novel thresholding technique to the Gaussian technique (it seems that these statistics are only shown for case studies within the scan subsets in Figs 16-19; if this isn't the case then clarification is needed). Since only lidar scans are used to compare success in wake characterization between methods, it's questionable if the LES case study is really adding anything to the manuscript.

**Major Comments:**

1. It seems that there are only one or two examples shown for output from the ATS method using LES. I understand the LES simulation is used to test the method on an idealized case but unless you're working with a larger subset of LES scans it doesn't seem relevant to put LES in the title of the manuscript? I also wonder if you even really need the LES simulation for this study since it just seems to be used to explain the methodology.

2. The LES scan case highlights an interesting example – the ATS algorithm results in many wake shapes (noise) in the far wake; when comparing the ATS v. threshold deficit methods, are you considering these far wake shapes in the confusion matrices? It's hard to tell if you completely ignored the far wake in the analysis or if it's included but acknowledged as a source of uncertainty? Because it seems like these shapes would affect calculation of the wake centerline when comparing ATS centerline to Gaussian centerlines?

3. The manuscript could benefit from better figure organization (some of the figures appear before they are called or are shown in incorrect sections, more on this in 8.). It seems the equations in section 4.1 might be somewhat out of order; it's unclear where WP is actually used in the process of the ATS method?

4. It would be interesting (and likely important) to know how the radial velocity values are transformed from radial to cartesian components.

5. The word "threshold" is used many times throughout the manuscript (i.e. in the introduction) before it's completely clear what type of threshold is really being applied; it would help to clarify this for a broader scope of readers.

6. It might be helpful at the end (or beginning) of the methodology to have a table outlining and summarizing all of the techniques you've used, the amount of data (and what type of data) you've used for each technique, and to also clearly show which techniques you are comparing. You're using two common techniques for comparison, comparing manually detected shapes and

ceterlines, and also switching up the datasets for some of them (i.e. the inclusion of the LES case study) so it is rather hard to follow. In the first paragraph of the Results section you broadly summarize your process but I don't think it's enough to help with the organization. Maybe even call these comparisons Experiment A, B, etc. and label that in the table and section headers? Just for ease of organization.

7. Do you have ensemble statistics for comparing Gaussian and ATS methods for centerline detection? As I understand it, Figures 16-19 are interesting case studies but the overall success of ATS v Gaussian is still ambiguous. Maybe you could have only one figure showing the R^2s for each wake (because it's helpful for visualization) and then have a table for ensemble statistics (so you would reduce your figure count by about 3).

8. Figures, figure labels and captions should be clarified:
   Figure 4 could use a label for the intensity colorbar below, and the caption states a "valid wind speed range" but this range isn't clear numerically.
   Figure 5 could use a label for the intensity colorbar below and labels for the entropy figures (a) and (c). It might also be helpful to put the location of turbines in this image as you did in other figures farther below.
   Figures 7 and 8 and Table 1 should be in the previous section.
   Labels for Figure 10c are ambiguous – what are helper lines?
   Figure 11 is quite hard to follow in general, and it's the first time the colorbar has been used to distinguish different wake shapes. This is another instance where discussing the output of the ATS (in terms of producing multiple wake shapes) would be helpful. Do you need the black dotted centerline in (d)? It might be helpful to switch (c) and (d) and remove the black centerline in d, also maintain a red color outline for the wake in (c) and (d) so readers don't think you are using different methods to get the red and black wake outlines.
   Figure 15 – I'm not sure why the corrupted scans are included in this figure since they were difficult to inspect manually?
   Table 2 should be in the previous section
   Figure 19 – is the "missing" gray color referring to the non-filled gaps?
   Figure 20 – not sure what invalid/valid and WIND and WAKE=WIND mean? Could you clarify?

9. The conclusion feels rushed and some of the statements in the conclusion about success rates of the Gaussian v. ATS in the near wake seem confusing / misleading – are you using ATS to identify centerline for far wake objects? Again it'd be helpful to have ensemble statistics for Gaussian v. ATS and better clarification as to whether or not far wake objects are included in success rate analysis.

**Minor Comments**

1. **Line 174** The word "significant" shouldn't be used unless significance testing has been done
2. **Line 125** The reference wind direction is mentioned several times and is an important part of the manuscript but it's unclear whether this is wind direction as given by the met mast?
3. **Line 141** Why did you choose the wind speed of 7 ms$^{-1}$ as appropriate?
4. **Line 105** It's a bit hard to follow when the reader has to scroll many pages down to Table 1; is it worth not mentioning table 1 and the scan subsets in this sentence instead mentioning them later, possibly in the data quality section.

5. **Line 163** Directional entropy is a bit of a misleading term. I was thinking it would be entropy in wind direction measurements of some sort; is there a better term to use for this?
6. **Lines 163-185** This discussion would benefit from including numerical entropy values for the scans discussed, i.e. "The entropy calculated across the beam range highlights several lidar scans with an entropy decrease compared to other cases (*insert mean entropy values for these cases)"* – as is, the point of this paragraph is somewhat hard to follow although the figure is helpful; could we have a succinct sentence summarizing these results at the end of the paragraph marked by line 187? It would be helpful to include numerical values to further contextualize Table 1 as well. Also it would be important to clarify what constitutes "corrupted" measurements – is this based on SNR/CNR filtering? Or is it based on the number of data points that exceed the wind speed limit as you discuss later? Either way, it would be helpful to clarify what this means the first time it's mentioned.
7. **Line 189** Can you be clearer about the numerical value of the percentage of data points that exceed the wind speed limit? Since this is an important criterion for identifying "corrupted scans"?
8. **Line 190** Why are the corrupted scans split into 5 subsets / why is that relevant?
9. **Lines 195-214** These should refer to Figure 8, not Figure 7, right?
10. **Lines 220-221** Can we have more detail about the phrase "shown to reproduce the wake shape rather accurately"?
11. **Line 261** I think it's worth stating in the previous paragraph that the output of the thresholding will result in multiple different shapes (particularly in the far wake), since this isn't visible for the initial LES case (since it's continuous). It's an important result of the thresholding algorithm so I think it's important to discuss it thoroughly and to discuss it well before you show it in Figure 11, for example.
12. **Lines 324-326** You could probably remove this paragraph.
13. **Lines 143-144** How many points constitute a "larger cluster" or high wind speed values? So if gap filling is not performed, are there just missing data values within the scan? And these are thus visible in the scan image?
14. **Line 330** I'm not sure what "The deficit-based threshold is presented as the wind-speed threshold" means – can you clarify?
15. **Line 347** How many scans do you choose to demonstrate the methods' performance? Why do you choose them? Again might be helpful to put this in a table.
16. Section 5.2 title should include a mention of the wake deficit method since you're also comparing that with manual detection and ATS
17. **Section 5.2** Would it be helpful to refer to manual detection as "ground truth" – so we know that you are comparing the performance of ATS and wake deficit to that of the manual detection and producing the confusion matrices as such? This just goes back to ease of organization; maybe a table as mentioned previously would make this clearer.
18. **Table 2** To understand the results better, could you explain why the manual-deficit exhibits such a comparatively high rate of false positives?

---

## Author Response (AR1)

**Final response**

Maria Krutova

January 2022

**Revision summary**

This is a short summary of the most important revisions. The detailed response to the reviewer's comments is provided after this section.

**Major revisions requested**

1. We provide the code, sample data set and plots used for the visual evaluation. As we cannot yet release the data and code in progress publicly, the data is uploaded to Zenodo under restricted access.

2. Pseudo-code algorithms were added for the automatic threshold finding and centerline detection from the thresholded data.

3. Expanded and detailed the description of the novel wake identification and characterization algorithms.

4. Detailed the description of the LES setup and Gaussian method parameters.

5. An example of wake identification on the averaged LES wake is now using the instantaneous LES wake to provide a more complex case for the demonstration. Full demonstration of LES wake detection is moved into a separate section to split it from the extensive analysis of the ATS method performance on the liar data.

6. Added qualitative comparison between the Gaussian and ATS methods. The quantitative comparison was hindered because both methods do not perform well in similar parts of the wake but fail in different aspects. Since the Gaussian method cannot be taken as a 'true' detection, we evaluated whether each method could identify wakes and their centrelines. We provide the plots used for the manual evaluation as a supplement.

7. Section 5.6 initially focused on the possibility of separating near and far wake in the case of the bimodal histogram. Since the reviewers had asked about the specifics of the wake detection in the bimodal case, we corrected the subsection to address the comments instead of the near-far wake identification, which is rather situational and does not transfer well to other cases.

8. The conclusion section was re-written and expanded to cover the efficiency of the pre-processing method, the performance of the ATS method in comparison to manual and Gaussian wake detection.

9. We prepared two supplements to accompany the manual analysis mentioned in the article: the manual segmentation to identify wakes and plots used for the evaluation of Gaussian vs. ATS method performance.

10. The video demonstrating ATS wake identification is updated to include manual and deficit-based thresholding. Since it may require additional corrections before the final revision, we do not yet upload it to the recommended portal, but provide together with the graph supplement.

11. The sections were re-arranged to address the comments on article structure. See Table 1.

**Minor revisions**

1. Restructured Section 2 so that the reference wind speed and direction are mentioned early and explicitly.

2. Changed general mention of the image processing to more precise mention of the thresholding method – that is the core method used for the wake identification and prepare data for the wake characterization (centerline search).

3. Specified whether wake identification or wake characterization was carried out, where it was relevant.

**Additional revisions to improve article quality and transparency**

1. Parallel to the review process, we continued to improve the code to make it less locked-on the specific parameters and improve the general performance. This slightly affected the look of few plots in the results, although the conclusions derived from them remained the same.

2. **(Major)** The entropy plots showing an overview for the whole data set originally presented the entropy calculated for the normalized data. We decided that the original plot did not illustrate well how the entropy reflected the presence of the wakes in the lidar data before pre-processing, as it had been applied to the normalized values. The plots were replaced with the entropy calculated for the original radial wind speed data, and the paragraphs describing the plots were corrected to reflect the observed patterns.

3. **(Major)** The alternate thresholding method (referred to as the 'deficit-based' method) based on the wake deficit had used a value of 95% of the free-flow wind speed to define the threshold. Initially, we took the reference wind speed measured at the cup anemometer as the free-flow wind speed. However, the lidar scans contain only radial wind speed, which may deviate from the actual wind speed when the wind direction and scanned azimuths differ. The direction difference explained the considerable underperformance of the deficit-based thresholding on the aligned and parallel subsets. We introduced a simple wind field reconstruction by projecting the radial wind speed to the reference wind direction so that the reference wind speed could be used as the free-flow wind speed. Applying the correction had gradually improved the thresholding result for the parallel wakes and slightly – for the aligned wakes. The bimodal subset remained unaffected, as the radial and actual wind speeds there are close to each other.

Table 1: Changes in the section structure

| Old sections | New sections |
| --- | --- |
| 1. Introduction | 1. Introduction |
| 2. Site description and measurement setup | 2 Data description |
| 3. Data description | 2.1 Lidar and reference data |
| 3.1 Lidar measurements at *alpha ventus* | 2.2 Large-eddy simulation |
| 3.2 Data quality | 3 Lidar data pre-processing and classification |
| 3.3 Information entropy and data classification | 3.1 Data quality |
| 3.4 LES setup | 3.2 Information entropy and data classification |
| 4. Methodology | 4. Methodology |
| 4.1 Automatic threshold detection | 4.1 Wake identification using automatic threshold detection |
| 4.2 Wake detection using image processing techniques | 4.2 Wake characterization from the thresholded data |
| 4.3 Wake detection using the Gaussian method | 4.3 Wake identification and characterization using the Gaussian method |
| 4.4 Wake deficit threshold | 4.4 Deficit-based wake identification via thresholding |
| 4.5 Manual wake detection | 4.5 Manual wake identification and characterization |
| 5. Results | 5 Proof of concept: Wake identification and characterization from the LES data |
| 5.1 LES wake detection | 6 Results |
| 5.2 Comparison of the ATS detection against the manual detection on lidar data | 6.1 Comparison of the ATS wake identification against the manual identification and deficit-based thresholding |
| 5.3 Centerline detection | 6.2 Comparison of the wake characterization using the Gaussian and ATS methods |
| 5.4 Comparison of the Gaussian and ATS methods | 6.3 Wind and wake direction |
| 5.5 Wind and wake direction | 6.4 Wake identification in the bimodal subset using the ATS method |
| 5.6 Near and far wake separation with a threshold | 7 Conclusions |
| 6 Conclusions | Appendix A: Image data processing |
| Appendix A: Image data processing | |

Since the deficit-based thresholding now uses different input data than the manual thresholding and the ATS method, the thresholds could not be compared anymore. The scatter plot comparing the thresholds against the manual threshold was replaced with the box plot showing the variation of true negative and true positive percentages in each subset type.

4. **(Minor)** line 256:

   *We fit a polynomial function with the order of five, since the corresponding fit returned low root-mean-square error (RMSE) while not altering the inflection point location significantly.*

   was changed to

   *We fit a function $F(k) = a_1 + a_2/k^5$ since the corresponding fit returned a low root-mean-square error (RMSE) while not strongly altering the inflection point location.*

   The previous definition erroneously called the fitting function 'polynomial', while it was one of the discarded options for the fit. The actual expression was also added for transparency.

5. **(Minor)** lines 303-305: removed expression for the double Gaussian function since it is not actually utilized in the article.

6. **(Minor)** line 483: *The discrepancy for the bimodal subset could be possibly reduced because of the longer wakes and, consequently, more precise regression fit.*

   It was unclear which fit was implied here: the regression fit to determine the wake direction or the fit to determine the offset between wind and wake direction. It should be the former.

   Changed to

   *The variation of wake direction in the bimodal subset (reference wind direction $140-160°$) was possibly reduced because of the longer wakes and, consequently, more precise regression fit to estimate the wake direction.*

7. **(Minor)** It was brought to our attention, that the sensors at FINO1 recorded time in UTC after May 14, 2016 instead of the local time. The time stamp in Fig. 2 description was corrected to reflect this.

**Reply to the reviewer comments 1 (RC1)**

Thank you for your comment. It was very helpful for revising the article structure and clarifying the parts that are not very common in wind energy studies.

**RC1: It seems that there are only one or two examples shown for output from the ATS method using LES. I understand the LES simulation is used to test the method on an idealized case but unless you're working with a larger subset of LES scans it doesn't seem relevant to put LES in the title of the manuscript? I also wonder if you even really need the LES simulation for this study since it just seems to be used to explain the methodology.**

The LES simulation serves as a simplified example to demonstrate the core methodology. We admit that the title can be misleading. The title was corrected, and LES-related examples

are moved into a stand-alone section, "Proof of concept", to separate them from the results. The LES examples now focus on the instantaneous wake instead of the averaged case.

Old title:
*Development of an image processing method for wake meandering studies and its application on data sets from scanning wind lidar and large-eddy simulation*

New title:
*Development of an automatic thresholding method for wake meandering studies and its application on the data set from scanning wind lidar*

**RC1: The LES scan case highlights an interesting example – the ATS algorithm results in many wake shapes (noise) in the far wake; when comparing the ATS v. threshold deficit methods, are you considering these far wake shapes in the confusion matrices? It's hard to tell if you completely ignored the far wake in the analysis or if it's included but acknowledged as a source of uncertainty? Because it seems like these shapes would affect calculation of the wake centerline when comparing ATS centerline to Gaussian centerlines?**

The far wake was always considered in the threshold verification. However, the current centerline detection algorithm processes only the first continuous wake shape, so the centerline detection from the ATS-detected wakes stops for some scans while the Gaussian method can continue (that does not indicate that it will succeed, though). We acknowledge the current limitation of the ATS centerline search but plan to improve it further.

**RC1: The manuscript could benefit from better figure organization (some of the figures appear before they are called or are shown in incorrect sections, more on this in 8.). It seems the equations in section 4.1 might be somewhat out of order; it's unclear where WP is actually used in the process of the ATS method?**

The figure and table positions are mainly caused by the LaTeX formatting. Since the corrections affecting the paragraph length could change a figure position, we left it for the final editing.

WP matrix is now referenced in the algorithm description.

**RC1: It would be interesting (and likely important) to know how the radial velocity values are transformed from radial to cartesian components.**

This misunderstanding could be confusion caused by the wording. The lidar data used in the thresholding are always stored in the rectangular matrix defined by the polar coordinate system (beam range, azimuth). This matrix is processed both by the thresholding and centerline algorithms. However, if plotted as it is, the polar matrix distorts the scanning field in the far and near lidar range. Hence, we converted the coordinate field from polar (beam range, azimuth) to Cartesian (x,y) in order to plot a scanned sector in a more human-readable way.

The thresholding via ATS always uses the radial velocity because this method does not strongly depend on the actual wind speed values but mostly on their relative value.

**RC1: The word "threshold" is used many times throughout the manuscript (i.e. in the introduction) before it's completely clear what type of threshold is really being applied; it would help to clarify this for a broader scope of readers.**

The threshold is now explained in the introduction when it is first mentioned:

*In the simplest case, a threshold is a value that splits the range into two parts: all values below the threshold fall into one group, while the values higher than the threshold form the second group. When applied to the wind field for the wake identification, a threshold would split the data into the wake and free-flow points.*

**RC1: It might be helpful at the end (or beginning) of the methodology to have a table outlining and summarizing all of the techniques you've used, the amount of data (and what type of data) you've used for each technique, and to also clearly show which techniques you are comparing. You're using two common techniques for comparison, comparing manually detected shapes and ceterlines, and also switching up the datasets for some of them (i.e. the inclusion of the LES case study) so it is rather hard to follow. In the first paragraph of the Results section you broadly summarize your process but I don't think it's enough to help with the organization. Maybe even call these comparisons Experiment A, B, etc. and label that in the table and section headers? Just for ease of organization.**

Thank you for the suggestion. A summary table is added to the end of the Methodology section, listing the method names, properties, and input data.

**RC1: Do you have ensemble statistics for comparing Gaussian and ATS methods for centerline detection? As I understand it, Figures 16-19 are interesting case studies but the overall success of ATS v Gaussian is still ambiguous. Maybe you could have only one figure showing the $R^2$s for each wake (because it's helpful for visualization) and then have a table for ensemble statistics (so you would reduce your figure count by about 3).**

The direct comparison between Gaussian and ATS methods is complicated because the Gaussian method does not perform well enough on the noisy scans and merging wakes to be considered a good reference. $R^2$ statistics worked when the manual centerline was presented as a 'true' value.

We expanded Sec. 5.2 (now – Sec. 6.2) with a comparison based on visual inspection: score the success rate of the Gaussian and ATS methods in the near and far wake. It highlights the cases of wake-wake interaction that were challenging for both methods and shows a good agreement in the performance.

Since this is another case of the manual analysis, we had prepared a supplement containing all plots used in the visual inspection.

**RC1: Figures, figure labels and captions should be clarified:**
**RC1: Figure 4 could use a label for the intensity colorbar below, and the caption states a "valid wind speed range" but this range isn't clear numerically.**

'Grayscale intensity' label was added.

The mention of 'valid speed range' was removed as it is not very relevant for the plot. The caption now reads as follows:

*Figure 4. Consecutive lidar scans from the bimodal subset. No outliers are present in scans (a) and (c), while scan (b) has wind speed spikes near the wind turbine position and in the far range; subplot (d) shows the intensity distribution for the same scans.*

**RC1: Figure 5 could use a label for the intensity colorbar below and labels for the entropy figures (a) and (c). It might also be helpful to put the location of turbines in this image as you did in other figures farther below.**

'Grayscale intensity' label was added to the colorbar. 'Radial entropy' and 'azimuthal entropy' labels were added to y- and x- axes of the corresponding plots to mark a direction in which the entropy was calculated.

**RC1: Figures 7 and 8 and Table 1 should be in the previous section.**
**RC1: Table 2 should be in the previous section**

The positions are adjusted as much as automatic formatting allows.

**RC1: Labels for Figure 10c are ambiguous – what are helper lines?**

This was an error in the plotting script; the 'intersection' labels should refer to the black dots. 'Helper lines' are labeled correctly and refer to the concentric circles described in the centerline detection algorithm.

**RC1: Figure 11 is quite hard to follow in general, and it's the first time the colorbar has been used to distinguish different wake shapes. This is another instance where discussing the output of the ATS (in terms of producing multiple wake shapes) would be helpful. Do you need the black dotted centerline in (d)? It might be helpful to switch (c) and (d) and remove the black centerline in d, also maintain a red color outline for the wake in (c) and (d) so readers don't think you are using different methods to get the red and black wake outlines.**

Figure 11 is changed to show different methods in two different subplots + overlaid wake contours and centerlines in the third subplot. This allowed keeping the same legend without color conflict. The color-coded wake shapes are now moved to Fig. 10.

**RC1: Figure 15 – I'm not sure why the corrupted scans are included in this figure since they were difficult to inspect manually?**

Some of the corrupted scans allowed to perform manual detection. We agree that it adds unnecessary complexity to the plot. Due to the corrections in the methodology for the deficit-based threshold, the thresholds cannot be compared directly anymore. This plot became obsolete and is replaced with a box plot summarizing the percentage of true positives and true negatives for different methods against manual thresholding.

**RC1: Figure 19 – is the "missing" gray color referring to the non-filled gaps?**

The 'missing' label is replaced with 'no data' in all relevant plots to improve clarity.

**RC1: Figure 20 – not sure what invalid/valid and WIND and WAKE=WIND mean? Could you clarify?**

The labels were dictated by the brevity to leave more space for the actual plot. The labels for the scatter plot were changed as follows:

- 'invalid' → 'corrupted'

- 'valid' → 'data'

WIND and WAKE referring to the respective directions and are labeled near the respective

axes.

**RC1: The conclusion feels rushed and some of the statements in the conclusion about success rates of the Gaussian v. ATS in the near wake seem confusing / misleading – are you using ATS to identify centerline for far wake objects? Again it'd be helpful to have ensemble statistics for Gaussian v. ATS and better clarification as to whether or not far wake objects are included in success rate analysis.**

The far wake centerline is detected by the ATS method when the detected wake shape is long enough. If the detected shape ends at, e.g., 4D, the ATS method cannot proceed, while the Gaussian method can – although it does not guarantee valid results for the Gaussian method. See the comment about ATS to Gaussian comparison.

The conclusion is now re-written to be more detailed on the methods used, their level of automation, advantages, and disadvantages.

**Minor Comments**

**Line 174 The word "significant" shouldn't be used unless significance testing has been done**

The word is removed, as well as other uses of it.

**Line 125 The reference wind direction is mentioned several times and is an important part of the manuscript but it's unclear whether this is wind direction as given by the met mast?**

The reference wind and direction were mentioned in lines 98-99. We agree that the mention was too brief and had to be more emphasized.

Section 2 is now rearranged to describe the site, reference data, and lidar scans in enough detail.

The reference data are now introduced as

*The FINO1 meteorological mast has a cup anemometer installed at 90 m above sea level and a vane installed at 100 m above sea level. The wind speed and direction measured with those instruments are used to characterize the free flow. We will further refer to them as the reference wind speed and direction, respectively.*

**Line 141 Why did you choose the wind speed of 7 ms-1 as appropriate?**

The line should be read as 'wind speed difference'. The value is chosen empirically for this particular data set.

**Line 105 It's a bit hard to follow when the reader has to scroll many pages down to Table 1; is it worth not mentioning table 1 and the scan subsets in this sentence instead mentioning them later, possibly in the data quality section.**

The early mention of Table 1 is removed.

**Line 163 Directional entropy is a bit of a misleading term. I was thinking it would be entropy in wind direction measurements of some sort; is there a better term to use for this?**

We chose this term to have a short reference to the Shannon entropy calculated across the

beam range or azimuth. Apparently, it matched a pre-existing term, which definition is different from what we had implied. The 'directional entropy' is replaced with the 'radial entropy' (earlier – 'entropy calculated across the azimuth') and 'azimuthal entropy' (earlier – 'entropy calculated across the beam range') to avoid confusion.

**Lines 163-185 This discussion would benefit from including numerical entropy values for the scans discussed, i.e. "The entropy calculated across the beam range highlights several lidar scans with an entropy decrease compared to other cases (insert mean entropy values for these cases)" – as is, the point of this paragraph is somewhat hard to follow although the figure is helpful; could we have a succinct sentence summarizing these results at the end of the paragraph marked by line 187? It would be helpful to include numerical values to further contextualize Table 1 as well. Also it would be important to clarify what constitutes "corrupted" measurements – is this based on SNR/CNR filtering? Or is it based on the number of data points that exceed the wind speed limit as you discuss later? Either way, it would be helpful to clarify what this means the first time it's mentioned.**

The entropy values are added. The paragraph now reads as

*For the scans with low noise, the entropy values fall into the range of 4-5 both in the azimuthal and radial direction. The entropy calculated in the azimuthal direction highlights several lidar scans with a substantial entropy decrease (Fig. 6a) – the value drops below two and tends to zero.*

The corrupted measurements refer to the non-physical values of the wind speed (100-1000 m/s) appearing in the scans due to the crosswind effects mentioned earlier in lines 115-130. The corrupted scans are now introduced as

*The scans taken near crosswind direction show a large number of non-physical wind speed values reaching 100-1000ms-1. We refer further to these scans as 'corrupted'.*

The entropy criteria are now described in more detail.

**Line 189 Can you be clearer about the numerical value of the percentage of data points that exceed the wind speed limit? Since this is an important criterion for identifying "corrupted scans"?**

The following lines were added: *The corrupted scans in the data set consistently have at least 1% of points exceeding [30 ms$^{-1}$] limit. The percentage drops to 0-0.05% for the rest scans and corresponds to the occasional spikes.*

**Line 190 Why are the corrupted scans split into 5 subsets / why is that relevant?**

The data set was initially split into subsets. The corrupted scans cover exactly five subsets and do not appear in other subsets. The total number of the corrupted scans is more relevant, indeed, so we remove the mention of the subsets. Their count would still be seen from the overview table.

**Lines 195-214 These should refer to Figure 8, not Figure 7, right?**
Corrected

**Lines 220-221 Can we have more detail about the phrase "shown to reproduce the wake shape rather accurately"?**
Details about PALM reproducing double wake and polynomial kernel of the model were

added with the emphasis on the aspect most interesting for the wake identification. Also added: more recent benchmark comparison of the different wind turbine models.

Line
*The results produced with the model were shown to reproduce the wake shape rather accurately (Vollmer et al. (2015)). Particularly, the model resolves the double peak in the wake deficit distribution near the rotor.*
was changed to
*he results produced with the model were shown to capture the reduction of the wake deficit with the downstream distance at the rate similar to the encountered for wind turbines (Vollmer et al., 2015, 2017; Doubrawa et al., 2020). The wake recovery aspect is particularly important to test the ATS method performance in the far wake. The currently used polynomial kernel also allows fitting the Gaussian function to compare it with the ATS method*

The mention of a double peak was removed, since it is not too relevant in the current article: we close holes caused by the double wake for the centerline search from the thresholded image, and start the Gaussian fitting after $1D$ where the double wake is not so prominent in the lidar data.

**Line 261 I think it's worth stating in the previous paragraph that the output of the thresholding will result in multiple different shapes (particularly in the far wake), since this isn't visible for the initial LES case (since it's continuous). It's an important result of the thresholding algorithm so I think it's important to discuss it thoroughly and to discuss it well before you show it in Figure 11, for example.**

The LES section is now re-organized to demonstrate the wake detection on an instantaneous wake. Figures 9-11 were affected by the re-organization. Figure 9 preserves its order number, Fig. 10-11 are moved further and changed numbers to Fig. 12-13.

**Lines 324-326 You could probably remove this paragraph.**
The paragraph was removed.

**Lines 143-144 How many points constitute a "larger cluster" or high wind speed values? So if gap filling is not performed, are there just missing data values within the scan? And these are thus visible in the scan image?**

We performed gap filling only for 1-3 adjacent spikes removed, as they left enough points around to perform a simple interpolation without introducing uncertainty to the data. The whole preprocessing procedure aimed to clean the lidar scans from the obvious outliers and reduce their effect on the normalization, but not alter the original data too much.

The gap filling was not performed for the deleted points (values above 30 m/s). The thresholded images would show those points as 'missing' ('no data' in new notation). No distinction is made between originally missing data and values removed during the preprocessing.

**Line 330 I'm not sure what "The deficit-based threshold is presented as the wind-speed threshold" means – can you clarify?**

This is the threshold calculated as 95% of the free flow wind speed (taken from the reference wind speed measured by the cup anemometer at FINO1). Unlike the ATS method, it does not require the wind field to be scaled to [0,1] range, , but is applied to the wind speed values. Since we had found an error in the method implementation to the radial velocity, the paragraph is

removed, and the whole subsection on the deficit-based thresholding is re-written.

**Line 347 How many scans do you choose to demonstrate the methods' performance? Why do you choose them? Again might be helpful to put this in a table.**

We chose an example from each of the most represented non-corrupted subsets (parallel, aligned, bimodal) to demonstrate the method performance in each case and highlight the potential problems. Where possible, we tried to stick to ensemble statistics since the scan selection could be highly subjective. A supplementary video was provided to present the results of the wake identification through thresholding for the whole data set.

**Section 5.2 title should include a mention of the wake deficit method since you're also comparing that with manual detection and ATS**

The subsection title is changed to '*Comparison of the ATS wake identification against the manual identification and deficit-based thresholding*'.

**Section 5.2 Would it be helpful to refer to manual detection as "ground truth" – so we know that you are comparing the performance of ATS and wake deficit to that of the manual detection and producing the confusion matrices as such? This just goes back to ease of organization; maybe a table as mentioned previously would make this clearer.**

Manual detection is now emphasized as the 'true' detection in the methodology subsection (Sec. 4.5).

We perform a manual segmentation to select an optimal threshold for each lidar scan and use it as a 'true' identification.

**Table 2 To understand the results better, could you explain why the manual-deficit exhibits such a comparatively high rate of false positives?**

Due to the improved implementation of the deficit-based threshold, the table is different now. The remark is still relevant for the selected subset, although the difference is not so dramatic now. Since the deficit-based wake identification defines a threshold through a fixed relationship of 95% of the reference wind speed, the method does not perform self-adjustment depending on the data quality. It appeared that the deficit-based threshold is underestimated compared to the manual thresholding and identifies more points as wakes while they actually belong to the background flow (false positives).

**Reply to the reviewer comments 2 (RC2)**

Thank you for your comment and very detailed review of the methods used. Because of one of the questions, we were able to identify an error in the deficit-based threshold implementation. I will answer it in the beginning since it had most affected the revision.

**L329: How is the "free flow" wind speed calculated? Is this automatic or manual?**

The free-flow wind speed is taken from the reference wind speed series since the lidar scan is quite noisy to rely on the specific value near a wind turbine. However, as explained in Sec. 3.2, the lidar measures radial velocity but not the actual wind speed. Thus, the threshold calculated

as 0.95Uref would not match the free-flow radial velocity well when the wind direction is too different from the scanned azimuths. This explains the better performance of the deficit-based threshold on the bimodal case (wind blows along the scanned azimuths; the radial velocity is rather close to the actual wind speed) compared to aligned and parallel wakes (the direction difference is higher there).

This was not relevant for the ATS method, as it scales all information to the range of [0,1] regardless of the specific wind speed values. On the contrary, the deficit-based method has to work with the wind speed data and wind speed threshold defined in the same frame. Typically, a retrieval procedure should be carried out to reconstruct the wind field. Since we were not interested in the local flow direction, estimating a wind speed would be enough. The radial wind speed $U$ is a projection of the actual wind speed vector $U'$ to the line of sight and can be approximated as:

$$U' = U/cos(\phi_{MET} - \theta)$$

where $\phi_{MET}$ is the wind direction (here – reference wind direction measured at FINO1) and $\theta$ is the azimuth.

The estimation is added to the subsection on the deficit-based threshold (adding it to the Data section would confuse whether the same procedure is applied during the ATS method or the Gaussian fitting – both of these methods rely on the original radial wind speed data).

Although being very crude, this estimation improved the performance of the deficit-based threshold for the parallel wakes, but not for the aligned subset – the threshold from the wind speed is still largely underestimated. The correction affected the results for the confusion matrices (Fig. 12-14) and former Table 2. The deficit-based threshold could not be compared to the manual threshold anymore, so we replaced Fig. 15 with a box plot to provide an overview on the thresholding methods performance.

The affected table (Table 2) containing subset statistics on detections is provided together with the new table (Table 3).

[Figure]

Figure 1: Scan #221 (aligned wakes subset), wake identification.

[Figure]

(a) **Old plot**             (b) **New plot**

Figure 2: Scan #60 (parallel wakes subset), wake identification.

[Figure]

[Figure]

(a) **Old plot.** Comparison of the manual-detected threshold with (a) the deficit-based threshold, and (b) ATS-based threshold through the whole data set.

(b) **New plot.** Ensemble statistics of true negative and true positive detections within the subsets.

Figure 3: Change of the ensemble plot to compare results of the thresholding methods.

Table 2: Comparison of the thresholding methods' performance against the manual detection.

| Data type | Subset | Scans | Manual-deficit, % | | | | Manual-ATS, % | | | |
|---|---|---|---|---|---|---|---|---|---|---|
| | | | TP | FN | FP | TN | TP | FN | FP | TN |
| Parallel wakes | 3 | 51−75 | 100 | 0 | 39 | 61 | 80 | 20 | 2 | 98 |
| | 4 | 76−100 | 100 | 0 | 41 | 59 | 97 | 3 | 4 | 96 |
| | 5 | 101−125 | 100 | 0 | 51 | 49 | 91 | 9 | 4 | 96 |
| | 6 | 126−150 | 100 | 0 | 41 | 59 | 95 | 5 | 4 | 96 |
| | 7 | 151−175 | 100 | 0 | 67 | 33 | 98 | 2 | 3 | 97 |
| | 11 | 251−275 | 100 | 0 | 65 | 35 | 99 | 1 | 4 | 96 |
| | 12 | 276−300 | 100 | 0 | 54 | 46 | 93 | 7 | 2 | 98 |
| | 16 | 376−400 | 100 | 0 | 55 | 45 | 96 | 4 | 3 | 97 |
| Transitional | 17 | 401−425 | 100 | 0 | 54 | 46 | 87 | 13 | 0 | 100 |
| Aligned wakes | 8 | 176−200 | 100 | 0 | 64 | 36 | 98 | 2 | 1 | 99 |
| | 9 | 201−225 | 100 | 0 | 33 | 67 | 100 | 0 | 10 | 90 |
| | 10 | 226−250 | 100 | 0 | 52 | 48 | 98 | 2 | 3 | 97 |
| Bimodal | 17 | 426−450 | 99 | 1 | 8 | 92 | 82 | 18 | 0 | 100 |
| | 18 | 451−475 | 96 | 4 | 10 | 90 | 89 | 11 | 0 | 100 |
| | 19 | 476−500 | 100 | 0 | 22 | 78 | 96 | 4 | 4 | 96 |
| | 20 | 501−525 | 100 | 0 | 24 | 76 | 85 | 15 | 2 | 98 |
| | 21 | 526−550 | 100 | 0 | 28 | 72 | 90 | 10 | 5 | 95 |
| | 22 | 551−575 | 100 | 0 | 33 | 67 | 90 | 10 | 8 | 92 |
| | 23 | 576−600 | 98 | 2 | 9 | 91 | 94 | 6 | 4 | 96 |

Table 3: Comparison of the thresholding methods' performance against the manual wake identification

| Data type | Subset | Scans | Manual-deficit, % | | | | Manual-ATS, % | | | |
|---|---|---|---|---|---|---|---|---|---|---|
| | | | TP | FN | FP | TN | TP | FN | FP | TN |
| Parallel wakes | 3 | 51−75 | 55 | 45 | 5 | 95 | 80 | 20 | 2 | 98 |
| | 4 | 76−100 | 69 | 31 | 4 | 96 | 97 | 3 | 4 | 96 |
| | 5 | 101−125 | 76 | 24 | 9 | 91 | 91 | 9 | 4 | 96 |
| | 6 | 126−150 | 85 | 15 | 9 | 91 | 95 | 5 | 4 | 96 |
| | 7 | 151−175 | 96 | 4 | 23 | 77 | 98 | 2 | 3 | 97 |
| | 11 | 251−275 | 95 | 5 | 22 | 78 | 99 | 1 | 4 | 96 |
| | 12 | 276−300 | 71 | 29 | 7 | 93 | 93 | 7 | 2 | 98 |
| | 16 | 376−400 | 80 | 20 | 6 | 94 | 96 | 4 | 3 | 97 |
| Transitional | 17 | 401−425 | 93 | 7 | 19 | 81 | 87 | 13 | 0 | 100 |
| Aligned wakes | 8 | 176−200 | 99 | 1 | 28 | 72 | 98 | 2 | 1 | 99 |
| | 9 | 201−225 | 100 | 0 | 13 | 87 | 100 | 0 | 10 | 90 |
| | 10 | 226−250 | 100 | 0 | 23 | 77 | 98 | 2 | 3 | 97 |
| Bimodal | 17 | 426−450 | 88 | 12 | 2 | 98 | 82 | 18 | 0 | 100 |
| | 18 | 451−475 | 83 | 17 | 1 | 99 | 89 | 11 | 0 | 100 |
| | 19 | 476−500 | 97 | 3 | 5 | 95 | 96 | 4 | 4 | 96 |
| | 20 | 501−525 | 97 | 3 | 7 | 93 | 85 | 15 | 2 | 98 |
| | 21 | 526−550 | 99 | 1 | 15 | 85 | 90 | 10 | 5 | 95 |
| | 22 | 551−575 | 100 | 0 | 20 | 80 | 90 | 10 | 8 | 92 |
| | 23 | 576−600 | 85 | 15 | 2 | 98 | 94 | 6 | 4 | 96 |

**Major Comments**

**This study is not replicable in its current state**

**The manually segmented wakes are not uploaded to a publicly accessible database. This is problematic, as two scientists would manually segment the scans differently, and as such, the results here are not immediately reproducable.**

That is a valid remark. We prepared the manual data set for the upload as well as the manual thresholds. The supplementary video is re-recorded to include manual wake identification and deficit-based thresholding results. Currently, we provide it together with the supplement containing the plots on manual thresholding and Gaussian vs. ATS comparison. If no further corrections are required, the video will be published on the AV Portal of TIB Hannover, as recommended by the publisher.

**L364-366: Additionally, parts of the novel algorithm are not presented in the paper.**

We added the most important parts of the algorithm (thresholding and centerline detection) as pseudocode to Sec. 4.1-4.2. The complete code is not entirely optimized yet and lacks a procedure to allow centerline detection in case of many disconnected shapes, so it would be available upon request for now.

**In its current state, the motivation for the study is somewhat weak**

**The new methodology seeks to improve the thresholding approach to wake identification. The authors state that "the most common wake detection method is. . . Gaussian". Is the thresholding approach common and important? When would this approach be used instead of the more popular Gaussian approach? The authors provide only one citation in the paragraph where the this technique is first discussed (L65), so it would appear that this approach is not commonly used.**

The thresholding approach was also used as a step in the analysis in Bastine, D.; Witha, B.; Wächter, M.; Peinke, J. Towards a Simplified DynamicWake Model Using POD Analysis. Energies **2015**, 8, 895-920. We did not include that reference, as their method required subtracting LES flow without wind turbine from the wake field, which is impossible to perform for lidar data. We added this reference to the introduction to show different applications of the thresholding approach.

We believe that the thresholding approach has limited use due to its low flexibility. The automatic algorithm would allow overcoming this disadvantage. Moreover, the algorithm is constructed in a way to require as little supplementary data as possible and thus can be used in the preliminary analysis.

**The validation methodology could be stronger**

**Wakes are manually segmented by hand-selecting a threshold between 0 and 1. I expected that wakes would have been segmented differently. I thought that a bounding countour was going to be manually drawn around every wake. This means that the validation study very specifically tests the question of "can we automatically pick the best threshold between 0 and 1" instead of "is our new idenfitication/characterization algorithm the best algorithm for the task". This is still presumably an important question, but we wish the authors to more clearly state that the primary goal of the text is to automatically pick the best threshold**

**for wake identification**

Yes, the comparison against manual detection describes whether the automated method can perform better than the fixed threshold and whether the ATS method performs at least on par with the manual thresholding. We chose this approach because it allowed performing comparison for the whole data set relatively fast. Drawing a bounding contour requires time similar to manual centerline detection; performing it for the data set even without the corrupted scans ( 475 valid scans) would be too time-consuming.

**The terminology regarding different wake identification algorithms was ambiguous, which made it difficult to assess what technique was being used at a given time**

"Image processing" is a massive field of study, and as such, it is ambiguous to refer to one wake identification algorithm as simply an "impage processing" technique. Please be more specific about this terminology, and if this technique is coming from the image processing field, provide a citation as such.

The wake identification uses thresholding methods – this will be more emphasized in the text whenever the image processing is mentioned. In the centerline detection algorithm, image processing is used only in the form of connectivity analysis and contour extraction. After the contour is extracted, the centerline detection mainly relies on the geometrical properties of the extracted contours.

**Throughout the manuscript, please make a stronger distinction when "wake identification" is being carried out vs. when "wake characterization" is being carried out, in accordance with the cited Quon et al. (2020). Unless its meaning is exceedingly obvious in the context, please refrain from using "wake detection"**

e.g. Sec 4.2: The title of this section says "Wake detection". However, this subsection appears to be describing wake *characterization* moreso than wake *identification* (and "detection" implies "identification").

Since we have only introduced a centerline detection algorithm but did not thoroughly analyze the wake behavior, it felt too much to call it wake characterization. Nevertheless, the point is valid; the descriptions are now corrected to refer to the thresholding method as 'wake identification' and centerline detection as 'wake characterization'. We also use 'centerline search algorithm' definition to reduce mentions of the detection in connection to the centerline.

**The structure of the different sections and subsections was unexpected, and as such, it made the narrative of the document more difficult to follow.**

I recommend that wake identification methodology subsections should be placed next to eachother in Section 4, and similarly, wake characterization metholody subsections should be grouped together

This suggestion is hard to implement because the Gaussian method performs both actions simultaneously. Splitting the manual detection would produce two short disconnected descriptions. As suggested by RC1, we summarized the methods in a table closing the Methodology section: the name by which the method is referred to further, input data, basic principles, and the automation level.

**I recommend that Section 2 be made into a subsection of the contents Section 3**

Section 3 already covers several different topics, so we do not feel it would benefit from

including technical information on the measurement site. We re-arranged Sect. 2 and 3, so that Sect. 2 contains only data description, while Sect. 3 describes the pre-processing and classification of the lidar data.

**Is this truly an automatic algorithm that avoids the need to manually preprocess and manually segment?**

Yes. All the corrections mentioned can be implemented and parametrized to run as a part of the main algorithm. This includes:

- Despiking and outlier removal depend on the parameters (see next comment). This procedure is not unique to the method and can be performed with any other tool convenient for the user.

- Selection of the wake shape to perform centerline detection: the algorithm either selects a shape that contains a wind turbine or if the wind turbine happens to lie in the free-flow, the largest wake shape in a prescribed radius (1-2$D$).

- Adjusting the wake shape for the centerline detection: if the wake shape contains more than one turbine, increase the threshold until wind turbines belong to different shapes. This is a typical case in the aligned wakes subset and some scans from the bimodal subset. The threshold adjustment is not crucial for the aligned wakes since the AV10 wake hits the downstream turbine directly, and the merged wake can be interpreted as one long wake structure. That is not the case for the bimodal subset, where two parallel wakes come close to each other. The thresholded wake shapes must be separated; otherwise, the centerline search algorithm may interpret two merged wakes as one wide wake and place centerline points incorrectly. Right now, we ran the thresholding algorithm with the same parameters for all scans. It could be that subset-specific processing and parameters (e.g., different curve fit to the derivatives during the threshold detection) can lead to a higher threshold, thus removing the necessity for the correction.

- Determination of the wake direction to avoid ambiguity during the centerline detection: if available, take the wind direction from measurements or run the centerline search method with a large step and approximate the wake direction from the found points.

The only non-automated part is the definition of the subset types, which was performed after the manual inspection of the data set and entropy characteristics. However, it was only used to structure the results of the wake detection. No subset-specific corrections were yet applied during the preprocessing and the wake detection.

**L139-144: Is the despiking process manual or automatic? If it is manual, that potentially hinders the ability to apply the ATS algorithm on larger datasets.**

It is automatic but uses two parameters: maximum allowed wind speed (here, radial wind speed magnitude) – values above are considered non-physical, and removed; and wind speed difference – if the wind speed falls into the allowed range but differs from the local mean of the nearby points by the specified value, it is considered a spike.

**Sec 4.1: This section has statements like "A similar point in the first derivative graph \*can\* be used" and "only a second derivative inflection point \*can\* be used". \*Are\* they used in your algorithm? Please be precise.**

Yes, we use both points since the threshold based only on the second derivative may appear too strict. The paragraph is re-written as follows:

*A similar point in the first derivative graph $T_1$ is used as a control value. We select the threshold as an average value between first and second derivative inflection points to smooth the threshold detection outcome. If the points initially laid close to each other, the averaged threshold $T = (T_1 + T_2)/2$ would not deviate too far from $T_2$. If the difference between $T_1$ and $T_2$ is high, the smoothing prevents the threshold from being too strict and leaving weak wakes undetected.*

**L265-266: The use of "preferably" makes me think that this algorithm is not automatic**

Knowing the wind direction prior to the centerline detection allows skipping few conditional steps in the centerline search algorithms. Suppose there is no information on the wind direction. According to the algorithm, we first draw a circle of a radius $1D$ centered at the wind turbine position.

(Note: the algorithm was updated and improved to cover more exceptions, so this description may differ in details compared to the original article)

If there are only two intersections, the algorithm calculates a middle point of the arc. Two intersections split a circle into two arcs, hence, two midpoints: 'upstream' and 'downstream' of the wind turbine. Two intersections guarantee that one point lies inside the wake shape – the algorithm can take it as a wake point and proceed to the next step by increasing the circle radius.

However, if the very first circle finds more intersections and the wind direction is unknown, the algorithm does not get enough information to select a centerline point. To allow it to continue, we first discard all midpoints in the free flow – that leaves only the points inside the wake shape. Then the algorithm draws few more circles with a step of 0.25-0.5D (the step is larger than is used for the centerline detection) and searches for new midpoints. All midpoints inside the wake are accepted at this stage since the actual wake direction is unknown. The wake direction is estimated by fitting a linear regression to the found points. While the wake direction estimated in a such way may strongly differ from the reference wind direction, it was generally enough to narrow the search for the actual centerline detection.

This complex algorithm is used solely to approximate the wake direction to resolve ambiguity in the centerline detection if no other information about wind or wake direction is provided. The information on the wind direction allows to skip to the estimation of the wake direction and reduce errors caused by the erroneous estimation.

The centerline detection algorithm restarts from the beginning with a smaller step ($\sim 0.1D$). Now, if two or more centerline point candidates are found, they are compared against the estimated direction (which midpoint inside a wake shape gives the lowest deviation from the estimated direction?) or local wake direction (which midpoint turns the wake by the lowest angle?). These checks attempt to prevent the centerline point identification upstream or on the side of an irregular wake.

**The conclusion does not sufficiently summarize the manuscript**

**Please summarize the strengths and weaknesses of the novel algorithm, especially relative to the other wake identification/characterization techniques**

The conclusion is now re-written to summarize the main features of the ATS method and areas for improvement. The uncommon pre-processing procedures such as entropy criteria are also detailed.

**Minor Comments**

**L3: What is the difference between a "wake pattern" and a "wake shape"?**
The terms were supposed to distinguish between a data set of dynamically changing wakes and an instantaneous wake. Since we are further referring to 'wake shapes' as a cluster of points detected by the thresholding method. Both mentions are replaced with more precise terms to avoid confusion.

Old:
*Scanning lidar data are used to identify the wake pattern behind offshore wind turbines but do not immediately reveal the wake shape. The precise wake identification helps to build models predicting wake behavior.*

New:
*Scanning lidar data are used to identify the wind flow behind offshore wind turbines but do not immediately reveal the wake edges and centerline. The precise detection of the wake edges and centerline helps to build models predicting wake behavior.*

**L18: It is a drastic oversimplification to state that the velocity deifict at 5D decreases to 20%. Consider citing review articles such as Stevens and Meneveua (2017) and Prote-Agel et al. (2020) in the introduction**
Line
*The relative velocity deficit rapidly decreases to 20% at the downstream distance of five rotor diameters (5D).*
changed and expanded into
*The wake region is characterized by decreased wind speed and increased turbulence intensity. The relative velocity deficit, or wake deficit, is strongest right after the wind turbine. Strongly affected by wind turbine rotor, the region extends up to $4-5$ rotor diameters depending on the terrain characteristics and stability conditions (Stevens and Meneveau, 2017; Porte-Agel et al., 2020). The wake transitions to the far wake, where the recovery to the free flow is considerably slowed down; at the same time, the wake width increases up to three rotor diameters according to observations (Aitken et al., 2014).*
**L20: Please provide a citation for "The typical turbine spacing in the wind farms is usually 8D", as I believe onshore and offshore spacing differ** The wind turbine spacing may largely even within a wind farm, e.g., London Array has spacing in the range of $650-1200\,\mathrm{m}$ making it $5.3-10D$ for the rotor diameter of $120\,\mathrm{m}$.

Since it hard to define a specific value, we change the line from
*The typical turbine spacing in operational wind farms is usually 8D...*
to
*The turbine spacing in operational wind farms reaches 7−10D (e.g., London Array)...*

**L38: Is it correct to say that lidar measurements are in situ? I think of lidars as remote sensing instruments**

Removed the mention of 'in situ'.

**L63: This sentence implies that thresholding algorithms are always applied 6–8D downwind of a turbine**

The paragraph

*The detection is applied up to a 6-8D distance downstream the turbine, where the wake structure remains primarily continuous.*

was changed to

*As shown by España et al. (2011), the method is effective for a regular flows, e.g., in a wind turbine: a threshold of 95% of the free-flow wind speed identified the continuous part of the wake up to the downstream distance of 6−8D.*

**L69: What is the difference between an "image" and "processed wind speed data"? Is an "image" a raw version of the wind speed data?**

The 'image' implies that the data did not originally have information about the wind speed. It could be a photo or a figure from the publication. We do not focus on images in the study but remark specifics in the Appendix.

The 'processed data' refers to the data after despiking and normalization by scaling to the range of [0,1]. While the value distribution resembles a respective grayscale image (as shown in Fig. A1), the data can be reverted to the original wind speed as long as the minimum and maximum wind speed values used in scaling are preserved.

Considering the comments on the use of the thresholding method
**L116-117: I am surprised to hear that you only encounter two types of noise. I imagine there are more variables that confound your signal (e.g. solid objects). Perhaps it is more appropriate to say "two primary challenges that obfuscate the wake signal" rather than "noise". Also, please clarify what is meant by "high wind speed due to the measurement error"**

Those types of noise were the types we had encountered in the particular data set and considered an obstacle for accurate wake detection.

The line
*Working with lidar scans, we encounter two types of noise: wind speed fluctuations not caused by the wake and high wind speed due to the measurement error.*

was expanded into
*Working with the current data set, we encountered two types of noise affecting the quality of the wake identification through thresholding: small wind speed fluctuations not directly caused by the wake and high wind speed values appearing due to a measurement error.*

The measurement errors due to crosswind effects and beam reflection from rotor blades were described further in the same subsection.

We placed a line

*The measurement errors are primarily caused by the difference between wind direction and lidar orientation.*

before the description of lidar limitations to allow smoother transition between paragraphs.

**L125: Where does the reference wind direction come from? A sonic on the mast? Wake angles?**

**L150: What does "reference data" refer to?**

The reference wind and direction were mentioned in lines 98-99. We agree that the mention was too brief and had to be more emphasized. Section 2 is now rearranged to describe the reference data prior to the more detailed description of lidar data.

The reference data are now introduced as

*The FINO1 meteorological mast has a cup anemometer installed at 90 m above sea level and a vane installed at 100 m above sea level. The wind speed and direction measured with those instruments are used to characterize the free flow. We will further refer to them as the reference wind speed and direction, respectively.*

**L163: What is "directional entropy"? How does it differ from Shannon entropy?**

We chose this term to have a short reference to the Shannon entropy calculated across the beam range or azimuth. Apparently, it matched a pre-existing term, which definition is different from what we had implied. The 'directional entropy' is replaced with the 'radial entropy' (earlier – 'entropy calculated across the azimuth') and 'azimuthal entropy' (earlier – 'entropy calculated across the beam range') to avoid confusion.

**L174-177: Are all the low entropy scans indicative of strong crosswind effects? The "also" on L175 makes it seem like this is only true sometimes.**

**Also to clarify, are "cross wind corrupted" scans and "spiked data" scans the same? Also, what is the difference between mostly blue scans (e.g. index 310) vs paritually blue scans (e.g. 405)?**

The entropy plot was presented for the data scaled to [0,1] – higher values are closer to zero, lower values are closer to one. The difference between 'mostly blue' and 'partially blue' is caused by the maximum radial velocity in a lidar scan. The corrupted scans tend to have clear non-physical values 100-1000 m/s, while spikes show more realistic values within 15-50 m/s. Scaling a range [0,1000] to [0,1] smooths low velocities stronger than when the same procedure is applied to a range [0,50].

We decided to replace the existing plot with the entropy calculated for the data before normalization to the range of [0,1]. It renders the same patterns at wind turbine positions but processes spiked and corrupted data differently from the entropy calculated for the normalized data. The corrupted scans can still be identified from the entropy across the beam range (similar to Fig. 6a), but the spiked scans blend with other non-corrupted scans.

**L182: Scans 1-50 and 301-375 show substantial decreases. Why are smaller ranges stated?**

The range for Fig. 6 was restricted to preserve the features for the non-corrupted scans. If the figure is plotted for the full range, the non-corrupted scans shift to red tones making the entropy fluctuations less distinguishable. At the same time, no interesting patterns are revealed for the corrupted scans except for the aforementioned gradual decrease of entropy. We decided

(a) **Old plot.** Entropy calculated for the normalized data. The colorbar range is limited.

(b) Entropy calculated for the normalized data. The colorbar shows full range.

[Figure]

[Figure]

(c) **New plot.** Entropy calculated for the original data. The colorbar range is limited.

(d) Entropy calculated for the original data. The colorbar shows full range.

Figure 4: Comparison of the entropy plots with a different colorbar range and input data.

to limit the color range to preserve more valuable information in the lidar scans suitable for analysis.

Figure 4 provided compares different presentations of the entropy plot and shows the new plot for the data prior to normalizaton.

**Figure 5: Label the AV7 and AV10 wake**
The labels were added.

**L210: Could you please clarify why the "bimodal subset" sees a bimodal distribution of wind speeds but the "parallel wakes" subset does not see a bimodal distribution? I don't understand why the "parallel wakes" subset also wouldn't see one large peak that represents the free flow and a second peak that represents the wakessds. Does this happen because the "bimodal" wakes largely stay within the field of view of the lidar whereas the "parallel" wakes leave?**
The two wakes in the bimodal case take a larger percentage of data points than in other cases. This happens because wakes form in the lidar near range. While the near range occupies a rather small area in m$^2$, it has the same amount of data points as the far range, which was

not subjected to the wakes in our data set. When the scan data are presented as a histogram, the far wake and free flow points are encountered nearly equally often in the bimodal subset – those are the points forming two separated peaks or merging into a flat one. Since the two long wakes do not regularly form within the scanned area, it limits the application of the thresholding methods requiring a distinctive bimodal histogram.

Appendix touches this property of the bimodal subset and shows how the far wake peak disappears when the histogram is plotted for the image of a lidar scan sector and not the original polar coordinates matrix. The reference to the Appendix is now added to the bimodal subset description.

Figure 2 is replaced with an example from the bimodal subset to show the difference between the lidar scan presented as a polar coordinate matrix and as a scanned sector plotted in the Cartesian coordinates.

Considering the other subsets: A hint on a double peak can be seen in some aligned subset scans, mostly because of good contrast between wakes and the free flow and large wake area in general. The parallel wakes subset has shorter visible wakes. The AV7 wake leaves the scanning area before the transition to the far wake. AV10 far wake is scanned at a higher height than AV7 (158 m vs. 97m, see Fig. 1c), so it is weaker and easier merges with the noise.

**Sec 3.4: What is the wind speed forcing of the LES?**
The description of LES input parameters is expanded, all parameters are now listed.
Old paragraph:
*The domain contains 2304×576×192 points and has the horizontal grid spacing of 4 m. The vertical spacing below 600 m is also 4 m, above 600 m the vertical spacing is stretched with a factor of 1.08, capped at maximum 8 m grid cell height. The reference NREL 5MW wind turbine has a hub height of 102 m and a diameter of $D_r$ =126 m and is placed in the center of the domain so that the wake length can reach up to $20D_r$. The surface temperature is 277 K and increases by 1 K per 100 m. Neither heat flux nor surface heating are activated. During the simulation the turbulence intensity reaches 6.6%.*

New paragraph:
*The domain contains 2304×576×192 points and has the horizontal grid spacing of 4 m. The vertical spacing below 600 m is also 4 m. Above 600 m, the vertical spacing is stretched with a factor of 1.08, capped at maximum 8 m grid cell height. The roughness length of $z_0 = 0.0005$ m corresponds to the calm sea surface. The Coriolis forcing is enabled for the latitutde of 54°, and the wind speed components are set to $u = 10.5$ ms$^{-1}$ and $v = -2.6$ m$^{-1}$ so that the flow rotation is compensated and the flow is aligned with the x-axis resulting in the horizontal speed of $10$ ms$^{-1}$ at the hub height. The surface temperature is 277 K and increases by 1 K per 100 m. Neither heat flux nor surface heating are activated. During the simulation the turbulence intensity reaches 6.6%.*

*The reference NREL 5MW wind turbine has a hub height of 102 m and a diameter of $D_r$ =126 m and is placed in the center of the domain so that the wake length can reach up to $20D_r$.*

**L232: You deal with at least two types of velocity distributions: unimodal (Fig 8a) and bimodal (Fig 8c). I am confused why you say you tend to only see one peak. As a reader at this point, I am wondering how the ATS algorithm performs on unimodal vs. bimodal data.**

The bimodal case has either two peaks of a different height or a single flat peak. This results in strong fluctuations for the second derivative of the intensity distribution. The first derivative shows one distinctive peak more often. The specifics of the bimodal subset are now covered in Sect. 6.4 (earlier – Sec. 5.6). The reference to the results from this section is also added to the methodology section.

**L248-257: You say "We detect the threshold at the point where... the curvature approaches zero". But when you also say that "the curvature graph tail may fluctuate and complicate the detection of zero curvature". I am confused - do you use curvature, the second derivative, or the first derivative to select your threshold. Are you doing this on polynomial-fit curves or on the raw curves? Please clarify. This is one of the most important sections of the paper, but it is difficult to understand your algorithm.**

We use first and second derivative curves to fit the function to find the concave point and estimate two thresholds. The final value is their average. We do not use the curvature plot because it can provide only one value for the threshold, which may be erroneous because of the fluctuations. The reasoning behind this is directly connected to the bimodal subset.

Usually, the second derivative in the bimodal subset has monotonically decreasing local minimums. Therefore, the minimum of the second derivative is likely caused by the free-flow peak. If the ATS algorithm incorrectly selects the local minimum and fitting range, the threshold could be overestimated and cut most of the far wake. Calculating a supplementary threshold from the first derivative and taking the average of two thresholds compensates for overestimation and preserves a larger part of the far wake.

**L264: What does "shape" mean?**

The term is now clarified at the end of the ATS method description as follows:

*Because of the wake irregularity, especially in the lidar scan, the method usually detects several clusters of high-intensity points. Any cluster may be a part of a wake as well as falsely detected noise. We do not yet distinguish between wake and noise and refer to all detected clusters as 'wake shapes'.*

The mentions of 'wake shape' as a general shape of the wake are now removed to avoid confusion.

**L269: Please demonstrate centerline detection on an instantaneous wake so we could get a sense of how this behaves on observational data**

The figures related to LES wake detection were modified. Figure 9 now demonstrates wake identification on the instantaneous LES data. Figure 10 shows the grayscale plot, thresholded image, and color-coded wake shapes. Figure 11 shows the wake characterization via the Gaussian method and centerline detection from the ATS method results. The contours and centerlines are plotted in a way to keep the uniform legend. An additional figure (new Fig. 10) shows a complex case of centerline detection for the irregular wake from a lidar scan to support the explanation of the centerline detection algorithm.

Figures 9-11 were affected by the section re-arrangement. Old and new figures are provided together with the updated captions as Fig. 5-7.

[Figure]

**Old plot.** Sample LES wake and threshold detection. (a) Original flow averaged over 10 minutes; (b) same flow normalized to the range of [0, 1]; (c) the intensity histogram of the normalized data; (d) CDF of the normalized data and CDF curvature; and (e) first and second derivatives of the CDF and the estimated thresholds.

[Figure]

**New plot.** Sample LES wake and threshold detection. (a) Original instantaneous flow; (b) same flow normalized to the range of [0, 1]; (c) the intensity histogram of the normalized data; (d) CDF of the normalized data and CDF plot curvature; and (e) first and second derivatives of the CDF and the estimated thresholds.

Figure 5: Updated **Fig. 9**. The averaged LES wake is replaced with the instantaneous wake to demonstrate the ATS method principles.

[Figure]

[Figure]

**Old plot.** Wake and centerline detection for a sample 10-minute averaged LES wake: (a) normalized flow field, same as Fig. 9b; (b) wake shape after threshold is applied; and (c) shape contour and centerline detection.

**New plot.** Wake and centerline identification for a sample instantaneous LES wake: (a) normalized flow field, same as Fig. 9b; (b) thresholded flow field; and (c) wake shapes color-coded to show connectivity.

Figure 6: Former **Fig. 10**, number changed to **Fig. 12** due to the section re-arrangement.

[Figure]

[Figure]

**Old plot.** Sample wake detection using idealized LES data. (a) Intensity histogram of an instantaneous flow field; (b) original data normalized to the range of [0, 1], (c) a thresholded image overlaid with the wake boundaries and centerline detected by the Gaussian method, and (d) color-coded wake shapes detected by the ATS method.

**New plot.** Sample wake identification and characterization using idealized LES data. (a) Thresholded data overlaid with the contour of the wake shape; (b) thresholded data overlaid with the wake boundaries and centerline detected by the Gaussian method; (c) ATS and Gaussian wake detection results, overlaid.

Figure 7: Former **Fig. 11**, number changed to **Fig. 13** due to the section re-arrangement.

**L276: Remove the word "presumably"**
Removed

**Fig 10c: "helper lines" and "intersections" have the same legend elements**
This was caused by an error in the plotting script; the 'intersection' labels should refer to the black dots. 'Helper lines' were labeled correctly and referred to the concentric circles described in the centerline detection algorithm.

**L283-284: Why is the wake direction ambiguous under these conditions? I would think that the wake direction is especially obvious in the "aligned" scenario**

The ATS algorithm often identifies AV7 and AV10 wake as a single shape in the aligned subset. While it is evident to a human eye, the algorithm running on the identified shape would detect a centerline point upstream and downstream for AV7 – both points looking equally valid for the algorithm because they lie inside a merged wake. If the wind or wake direction is known, the algorithm can only search for the centerline points close to that direction. If no particular search direction is prescribed, the algorithm cannot choose a centerline point of two (or more) unless special routines are run to approximate the wake direction before the thresholding.

**L285: I asked this earlier, but where does the reference wind direction come from?**

See earlier comment on the reference wind direction.

**L287-288: This seems like a major limitation. In the conclusion, please note that your algorithm does not work on weak wakes.**

These lines explained why we disregard AV11 in the analysis. The turbine was scanned at the lidar's far range and near the scan border. The AV11 wake is not seen in the 'parallel' subset due to the flow direction leading it outside the scan; the wake is often obstructed by the low-speed formations. However, it becomes more clear in the aligned and bimodal case. Still, the part of the wake captured is small. Overall, the weakness of the wake was not the primary problem for the identification, unlike the fact that very little information could be used from it compared to the other two wakes.

The line

*The wake from the wind turbine AV11 is usually weak and easily confused with the noise; we do not consider this wake in our analysis.*

was expanded into

*The wake from the wind turbine AV11 is prominent only for the bimodal subset and is too short and easily confused with the noise in the other subsets. We do not consider this wake in our analysis due to the little information it can provide compared to the other two wakes.*

The point on weak wakes is rather valid for the parallel subset. We briefly cover it in the comparison of different method results and emphasize in the conclusion.

**L295: "This feature" implies that the important feature is the transition from a single Gaussian to a double Gaussian. I believe you would like to say that the Gaussian distribution of wake deficit speeds is the important feature**

The line

*The wake deficit distribution is similar to the Gaussian distribution in the far wake (Ainslie (1988)) and often shows a double Gaussian peak in the near wake (Magnusson (1999)). This feature makes a base for a widely used method to detect wake boundaries and centerline (Vollmer et al. (2016); Krishnamurthy et al. (2017)).*

was changed to

*The wake deficit distribution is similar to the Gaussian distribution in the far wake (Ainslie, 1988) and often shows a double Gaussian peak in the near wake (Magnusson, 1999). The*

*similarity to the Gaussian distribution makes a base for a widely used method to detect wake boundaries and centerline (Vollmer et al., 2016; Krishnamurthy et al., 2017).*

**L297: Can you roughly quantify "small plane inclination"?**

About the same where we can use approximation $\sin \alpha \sim \alpha$. If a wind turbine is scanned at the hub height, the plane elevation of 0.1 rad (5.7°) will capture the wake until $5D$. Considering the wake expansion, the wake could also be captured at further downstream distances of $\sim 10D$.

On the second pass, we decided to remove this mention, since we do not research in details the performance of the Gaussian method depending on the elevation angle. The observed discrepancy between wake and wind direction posed larger problem for the Gaussian fit and was addressed in the article. The weak wakes in parallel subset and consequent underperformance of the method are caused by the scanning height itself, but not the scanning height changing along the wake. The scanning height changed strongly for the bimodal and aligned wakes subset but did not affect the fit quality.

**L319: Do reference wind directions and actual wake directions often significantly differ? If so, could you quantify a large discrepancy from either literature or this analysis?**

We added citation of Gaumond et al. (2014) in this section but leave the details for the discussion in Section 5.5, where this problem is covered. Gaumond et al. (2014) observed a normal distribution of cup anemometer measurements taken within 10 minutes with a standard deviation of 2.67°. We observed 5° average offset for the wake direction, which was enough to divert the automatic extraction of the wake profiles for the Gaussian method.

**L343: Is the centerline assumed to be a straight line (as is assumed with the Gaussian calculations) or it is allowed to turn?**

Both Gaussian and manual centerlines can turn. Some wording in the Gaussian section probably implied that we regard a straight line, while this is not the case. The Gaussian method returns one center point per wake profile. However, we need to extract the wake profile to get the data for fitting. This was done by extracting data along the line perpendicular to the wind direction. The Gaussian distribution peak may shift from the middle of the extracted profile depending on the wake movement. More details are added on the Gaussian fitting, and we cite Krishnamurthy et al., 2017, where the fitting was applied exactly to a lidar scan containing radial wind speed data.

The manual centerline does not require intermediate lines; we can draw it as a curve over the scan.

**L356: Do additional errors occur or do they not occur?**

The wording was changed and detailed to avoid confusion, additional changes were made due to the re-organization of the section.

Original:

*Figure 11b compares the centerline and wake shape detected by the Gaussian and ATS methods. Overall, the Gaussian method performs well in the range of $1 < x/D < 10$. Additional errors may occur in the far wake ($x/D > 10$), where the wake recovers to the free flow and the wake deficit function becomes too weak to fit accurately.*

New:

*Figure 13c compares the wake centerline and edges detected by the Gaussian and ATS methods. Both methods perform well in the range of $1 < x/D < 10$ and show good agreement on the same distance (Fig. 13c). Downstream ($x/D > 10$), the wake becomes weaker as it recovers to the free flow. If the wake deficit function becomes too flat to fit accurately, the fitting result may place the wake center incorrectly or overestimate the standard deviation and, consequently, the wake width.*

**L371: Does "wake deficit" refer to the Gaussian method?**

No, this is the method that uses a 5% wake deficit or 95% of the free flow wind speed as a threshold.

The line

*The confusion matrix $2 \times 2$ describes the comparison of the automatic thresholding methods (wake deficit or ATS) against the manual method and contains the following outcomes:*

was changed to

*The $2\times2$ confusion matrix describes the comparison of the automatic thresholding methods (deficit-based or ATS, see Table 2) against the manual method and contains the following outcomes:*

Table 2 is a new table that summarizes the methods used for wake identification and characterization.

**L387: What does "it" refer to?**

The manual threshold criteria. The line

*The aligned wakes subset utilizes the same manual threshold criteria for the wake splitting as bimodal subset (Fig. 13), although it may be harder to fulfill.*

was changed to

*The aligned wakes subset utilizes the same manual threshold criteria for the wake splitting as the bimodal subset (Fig. 15), although the condition may be harder to fulfill.*

**Figure 15: Why is the Corrupted data included here but excluded in Table 2?**

The original intention was to show data for the scans that allowed manual identification despite strong crosswind effects. The figure was removed after we had to correct the implementation of the deficit-based method, as the thresholds could not be compared directly anymore.

**L430: Could you please remind me - does the ATS centerline detection only work for the closest wake shape? Is that why the algorithm doesn't detect the centerlines in the far wake?**

Yes, we detect the centerline only for the first continuous shape near the wind turbine as of the current moment. Searching for the centerline in the far wake requires determining which of the disconnected shapes downstream are identified correctly as a wake and the order in which they should be connected. We tested an algorithm for the LES case, which used the scheme: search for the next large shape in the downstream wake direction – extract the centerline – recalculate the wake direction – repeat until the end of the domain is reached. This algorithm does not apply well to the lidar data due to the higher amount of false positives and a different way to store data. The LES data are simulated to align with one of the Cartesian axes, which gradually simplifies the centerline search. The lidar data are stored in the matrix defined by the polar coordinate system, i.e., the geometrical approach requires coordinate transformation

back and forth. We are now looking for an alternative solution for the centerline algorithm and see it as a good topic for a follow-up study.

**L450: The previous section also compared Gauss and the ATS. Please be more specific with this header.**

Former Sec. 5.3 and 5.4 are merged into one subsection. The titles are now changed from

*Section 5.2. Comparison of the ATS detection against the manual detection on lidar data*

to

*Section 6.1. Comparison of the ATS wake identification against the manual identification and deficit-based thresholding*

and from

*Section 5.4. Comparison of the Gaussian and ATS methods*

to

*Section 6.2. Comparison of the wake characterization using Gaussian and ATS methods*

The section numbers were altered due to the re-organization and introduction of a new section 'Proof of concept: Wake identification and characterization from the LES data'.

**L465-466: I am surprised that the estimated wake direction deviates so strongly from the reference wind direction. You cite a few studies in the following paragraphs. How large are the deviations in those studies?**

The deviations are not always mentioned in the references explicitly to quantify them. Gaumond et al. (2014) provides a $2.67°$ standard deviation for the measurements. From the plots provided in publications on LES simulations and lidar measurements, we see that the Coriolis effects become noticeable at $\sim 6D$. That could partially explain the difference between wind and wake direction because the distance between met mast and closest wind turbine is $\sim 8D$. However, that explanation might not be enough. We had observed an offset of an order of $5°$. If the same effect was observed for the wake, it would imply a deviation of the wake center $8D \cdot \tan(5°) = 0.7D$. The cited articles show weaker effects for about the same distance. Besides, we could expect a normal distribution of the deviations due to several effects (measurement error, yaw deflection, atmospheric conditions), yet we also see nearly constant offset.

Apart from the Coriolis effect, we are leaning towards the imperfection of the lidar orientation as the main cause for the deviation between wind and wake. We have performed a quick comparison to SCADA data for the same period and noticed a similar offset for the wind direction measured at the mast by the anemometer and at the wind turbine by the SCADA system – this supports the assumption that the wind-wake deviation is present and is not caused by the limitations of the centerline search algorithm or yaw deflection. Since the SCADA data deals with its uncertainties and was not a focus of the original study, we do not yet include it in the revised article and regard it as a good reference data set for the next study.

**L490: See my comment about L210. Also, does your quantification of "near wake" agree with standard definitions of "near wake"? Also, please state why someone would want to distinguish between "near wake" and "far wake" within a lidar scan.**

This section mainly served to show the capability of the ATS method for the particular case

of merging wakes. Based on your comment on the ATS method performance for unimodal vs. bimodal histogram, we re-wrote this part to focus more on the bimodal case specifics.

**L505-506: As written, a reader would not understand that you developed a new preprocessing methodology. Please make that clearer.**

The main novelty includes only entropy criteria. The despiking part is not specific to the method and can be performed in any other way convenient for the user. This will be emphasized more clearly.

---

## Referee Report (RR1)

'Development of an image processing method for wake meandering…'

by M. Krutova et al.

This is an interesting paper which deals with the difficult problem of reliably identifying wind turbine wakes within a generally turbulent flow field. The paper focuses on methods which use thresholding techniques applied to the image processing of Lidar data. A standard technique for identifying a wake from a set of scanned Lidar data of the velocity field, the Gaussian Wake Deficit method, uses a fixed threshold (usually 95%) of average free stream wind speed to fit a Gaussian cross-wake profile. Because of background turbulence and other noise in the signal this procedure has drawbacks causing errors. The present paper describes an alternative threshold-based method, a development of Adaptive Thresholding Segmentation (ATS) which has been used for other image processing problems such as identification of (sea) waves. In this method first and second derivatives of the cumulative PDF data is used to provide automatic threshold detection. Wake centre-lines are also determined. The new technique is applied first to numerically simulated turbulent flow fields containing turbine wakes generated by LES CFD, and then to Lidar data from the FINO1 tower covering three turbine wakes in the Alpha Venturus offshore windfarm. The wind-speed image data is processed using both the ATS method and the Wake Deficit method. Comparison of the results shows that the two methods are approximately equally successful and comparable in accuracy of detecting wake boundaries and centre-lines in the near wake regions behind rotors, but that in the far wake where accurate detection becomes more difficult the ATS method is significantly more reliable than the wake deficit method. The paper describes the process well and gives a good discussion of the difficulties and errors in the procedure. The code for the process is also available.

This a very useful paper and is suitable for publication in the Journal.